# HOW TO TRAIN DATA-EFFICIENT LLMS

**Noveen Sachdeva, Benjamin Coleman, Wang-Cheng Kang, Jianmo Ni**
**Lichan Hong, Ed H. Chi & Derek Z. Cheng**
Google DeepMind
`{noveen,colemanben,wckang,jianmon,lichan,edchi,zcheng}@google.com`

**James Caverlee**
Texas A&M University
`caverlee@cse.tamu.edu`

**Julian McAuley**
University of California, San Diego
`jmcauley@ucsd.edu`

## ABSTRACT

The training of large language models (LLMs) is expensive. In this paper, we study data-efficient approaches for pre-training LLMs, *i.e.*, techniques that aim to optimize the Pareto frontier of model quality and training resource/data consumption. We seek to understand the tradeoffs associated with data selection routines based on (i) expensive-to-compute *data-quality* estimates, and (ii) maximization of *coverage* and diversity-based measures in the feature space. Our first technique, ASK-LLM, leverages the zero-shot reasoning capabilities of instruction-tuned LLMs to directly assess the quality of a training example. To target coverage, we propose DENSITY sampling, which models the data distribution to select a diverse sample. Testing the effect of 22 different data curation techniques on the pre-training of T5-style of models, involving hundreds of pre-training runs and post fine-tuning evaluation tasks, we find that ASK-LLM and DENSITY are the best methods in their respective categories. While coverage sampling techniques often *recover* the performance of training on the entire dataset, training on data curated via ASK-LLM consistently *outperforms* full-data training—even when we sample only 10% of the original dataset, while converging up to 70% faster.

## 1  INTRODUCTION

Large language model (LLM) pre-training is perhaps the most data- and compute-intensive task attempted by the machine learning community to date, with impressive capabilities primarily being accomplished by training massive transformer architectures on trillions of tokens of text (OpenAI, 2023; Touvron et al., 2023b; Gemini et al., 2023; Comanici et al., 2025; Singh et al., 2025).

But even these incredibly capable LLMs are subject to empirical scaling laws, which predict sharply diminishing returns from a linear increase in model- or data-size (Hoffmann et al., 2022; Kaplan et al., 2020). Power-law scaling therefore acts as a soft limit on model quality, beyond which it is prohibitively expensive to drive performance by scaling up the data or model. At the same time, Sorscher et al. (2022)—in the context of vision pre-training—show that we can significantly improve the power law constants in the aforementioned scaling laws if we prioritize *important* training examples using some robust notion of data quality or impact.

A similar call for data-curation is also apparent in the context of training LLMs, where our largest models are quickly approaching their capacity and data thresholds. LIMA (Zhou et al., 2023) showed that LLaMA-65B (Touvron et al., 2023a) can be better aligned with human preferences when trained on a set of 1,000 carefully selected fine-tuning prompts, compared to training on as much as 52,000 unfiltered examples. Tirumala et al. (2023) recently conducted a large-scale data-efficient pre-training evaluation, showing that a 6.7B OPT model (Zhang et al., 2022) can converge up to 20% faster on data curated by a technique based on stratified cluster sampling. The Phi-2 experiments also suggest that when data curation is performed at a human-expert level (*e.g.*, by textbook editors), models can outperform baselines that are up to 25x larger (Javaheripi et al., 2023).

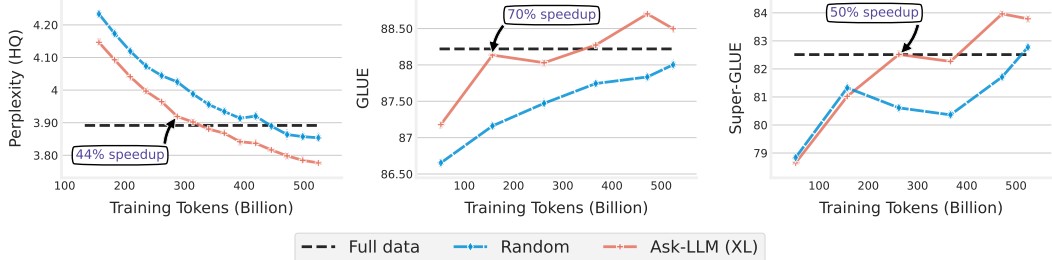

Figure 1: Data-efficient pre-training run of T5-Large (800M) using ASK-LLM with Flan-T5-XL as the data quality scorer. Training on 60% of the original dataset, ASK-LLM is able to train T5-Large both better and 70% faster, compared to training on 100% of the dataset.

Data curation routines can be fundamentally characterized as selecting training samples for quality, coverage, or some mixture of both (Figure 2). In this work, we seek to understand how quality and coverage affect the data efficiency of LLM pre-training. Our core research question is:

*"Are cheap-to-compute heuristics like maximum-coverage enough to pre-train a SoTA LLM, or are there real benefits from costly samplers that carefully evaluate the quality of each example?"*

This question is crucial to answer because data-curation algorithms can improve the Pareto frontier of the data-quantity↔model-quality tradeoff, directly addressing the bottleneck of power-law scaling by enabling higher-quality models to be trained using less data. Data curation also unlocks new tradeoffs between training time, inference cost, data collection effort, and downstream performance. For example, if we consider the compute-constrained (single-epoch) regime, a data-efficient LLM training routine may reach the desired performance using only X% of the data (corresponding to a $\leq$X% training speedup).

Despite considerable interest from the community for building data-efficient training methods (Sorscher et al., 2022; Paul et al., 2021; Coleman et al., 2020; Jiang et al., 2019; Katharopoulos & Fleuret, 2018), large-scale analyses of data pruning strategies are rare because of the extreme computational cost—especially in the context of LLM pre-training. To be more specific, an extensive comparative study necessarily entails pre-training (i) various sizes of LLMs, (ii) for a variety of data sampling rates, (iii) obtained through various pruning strategies. Further, downstream evaluations for LLMs also frequently involve fine-tuning, which is resource intensive in itself.

**Contributions.** We hypothesize that the roles of coverage and quality depend on the stage of training, size of the model, and the sampling rate. To understand the coverage/quality design choice better, we develop new data-efficiency routines that independently (and solely) target quality and coverage. Our ASK-LLM sampler prioritizes high-quality and informative training samples by asking a *proxy* LLM. Our DENSITY sampler seeks to maximize the coverage of latent topics in the input dataset through a diversified sampling procedure. To summarize, our contributions are as follows:

- **ASK-LLM sampling.** We develop ASK-LLM, a data curation technique that can train better models (*vs.* training on the *entire dataset*) even after removing up to 90% of training samples, while also consistently outperforming other well-established data curation routines. Furthermore, we find ASK-LLM also promotes data-efficiency during training (Figure 1).

- **Exhaustive benchmark.** We implement 22 different sampling strategies for pre-training T5-Large (800M) and T5-Small (60M) on 524B tokens and evaluate them on 111 downstream evaluation tasks. This leads to a total of 220 pre-training and 1,100 distinct fine-tuning runs.

- **New insights.** By analyzing the differences between ASK-LLM and DENSITY sampling, we study the role of coverage, quality, and sampling cost in LLM pre-training. We support our conclusions with additional studies of the convergence rate, correlations between sampler outputs, and impact of sampling cost on downstream performance. Our results show that while coverage sampling often *recovers* the performance of training on the full dataset, ASK-LLM (quality filtering) can often *exceed* it. These experiments suggest that LLM-based data quality raters are a worthwhile and effective way to drive performance in pre-training.

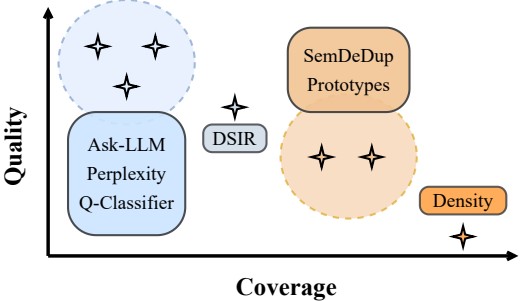

Figure 2: While there is no inherent tradeoff between coverage and quality, samplers target these metrics on a spectrum (up and to the left indicates a more aggressive prioritization). See Section D for a detailed description.

## 2 RELATED WORK

Data selection is a classical problem with well-established literature on coresets, sketching, importance sampling, filtering, denoising, and a host of other algorithms with similar goals. While we cannot possibly catalog the entire sampling literature, we hope to provide an overview of the principles behind common data selection algorithms. We also describe how these algorithms have been applied to machine learning, focusing on language model training.

**Coverage Sampling.** The first class of methods maximize the *coverage* of the sample by selecting points that are evenly distributed across the entire input domain, e.g., an $\epsilon$-net for a Lipschitz function (Phillips, 2017). Coverage sampling is motivated by the intuition that we ought to show a language model the full breadth of genres, topics, and languages (Longpre et al., 2023b).

Coverage sampling is typically accomplished by embedding examples into a metric space and selecting points which are mutually far from each other (Lee et al., 2023). A popular implementation of coverage sampling are cluster sampling algorithms, which groups inputs based on embedding similarity and selects representatives from each group. These algorithms are popular, scalable, interpretable, and enjoy strong theoretical support (Tukan et al., 2021; Feldman et al., 2020). However, there are also recent techniques based on submodular coverage optimization (Chen et al., 2012; Indyk et al., 2014; Borsos et al., 2020), models of the data distribution (Coleman et al., 2022), discrepancy minimization (Karnin & Liberty, 2019), and deduplication through token matching / similarity hashing (Lee et al., 2022).

Many variations of cluster sampling have been applied to vision and language model training. Sorscher et al. (2022) propose the "SSL prototypes" method for vision models, which removes points that fall too close to the nearest $k$-means centroid. SemDeDup (Abbas et al., 2023) also removes points based on this distance, but targets pairs of nearby examples, or "semantic duplicates," and prefers points close to the centroid. The D4 sampler chains MinHash deduplication, SemDeDup, and SSL prototypes together to prune both high-variance, sparse regions and prototypical, dense regions of LLM pre-training datasets (Tirumala et al., 2023). Coleman et al. (2020) considers a $k$-centers submodular selection routine on the last-layer embeddings of ResNet vision models.

**Quality-score Sampling.** Another class of methods are based on *quality scores*, where a scoring algorithm rates every example and the sampler preferentially selects points with high scores. For example, the selection-via-proxy (SVP) algorithm determines the importance of an input using the validation loss and uncertainty scores of a pre-trained model on the input (Coleman et al., 2020; Sachdeva et al., 2021). Paul et al. (2021) sample according to an "EL2N score" formed by ensembling the losses of 10 lightly-trained models. Ensemble prediction variance has also been used as the scoring metric (Chitta et al., 2021), as have ensemble disagreement rates (Meding et al., 2021). Other scores measure whether an example is likely to be forgotten (Toneva et al., 2019), memorized (Feldman & Zhang, 2020), or un-learnable (Mindermann et al., 2022).

In the context of pre-training LLMs, perplexity-filtering is one of the arguably most used quality-scoring technique, which prioritizes samples with *low* perplexity or conversely filters out highly

surprising examples (Wenzek et al., 2019; Marion et al., 2023; Muennighoff et al., 2023). Notably, recent advancements in cheaper to run model-based *training-run simulators* for LLMs can be used to *estimate* the perplexity of a training sample instead of running an LLM inference (Guu et al., 2023). Another group of methods selects training data that minimizes the *distance* between the distribution of selected data and a handcrafted high-quality data source (typically wikipedia and books). Typical ways are to do this in a feature space (Xie et al., 2023b) or by training a contrastive-style classifer (Radford et al., 2019; Anil et al., 2023; Javaheripi et al., 2023). Similar ideas have also been explored for optimizing the data mixture weights for pre-training (Xie et al., 2023a).

More recently, the community has decomposed the LLM data efficiency problem into quality-based data filtering and data mixing (Li et al., 2024). There are a variety of methods for each setting, including score-based re-weighting of the corpus (Kamath et al., 2025; Shukor et al., 2025) and aggregation of multiple rubric-based scores for filtering (Wettig et al., 2024). Data quality scores can also be derived from the evaluation results of small proxy LLMs trained on data subsets (Diao et al., 2025), and there are recent attempts to combine quality and diversity measures (Liu et al., 2025).

## 3  METHODS

We propose two samplers, ASK-LLM and DENSITY. These samplers have significantly different costs— ASK-LLM requires an LLM inference call for each training sample, whereas DENSITY is based on a diversified sampling routine that is cheaper than even clustering the dataset. They also exhibit substantially different selection behavior: ASK-LLM conducts a highly *nuanced* and *contextual* quality evaluation for each sample, while DENSITY asks whether we have already sampled many similar examples. By studying samplers on extreme ends of this spectrum, we hope to better understand the salient factors for LLM data curation.

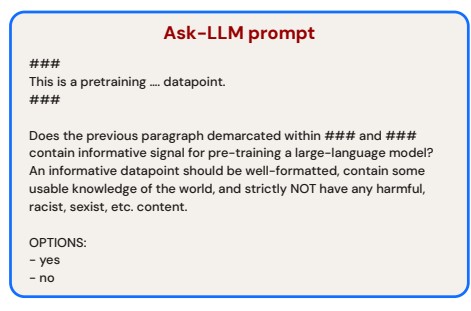

Figure 3: The ASK-LLM prompt to obtain each example's sampling score.

### 3.1  ASK-LLM SAMPLING

**Intuition.** Our intuition is that humans can easily identify commonly occurring failure modes in state-of-the-art data quality scorers. Hence, it should be possible to correct these mistakes using the reasoning capabilities of modern instruction-tuned LLMs. To do so, in ASK-LLM, we directly prompt an instruction-tuned *proxy* LLM with the prospective training example for the sampling decision (Figure 3). We take the softmax probability of the token "yes" as the estimated data-quality score. This procedure avoids the following common failure modes of perplexity filtering (*i.e.*, keeping the lowest perplexity examples). See Section H) for a qualitative analysis.

**Contextuality.** Perplexity filters often select samples that lack context, *e.g.*, containing questions without answers (Examples 11, 12, 15). ASK-LLM correctly identifies that these examples do not provide new information.

**Nonsense.** Perplexity filters can select examples that repeat common words / phrases (Examples 14 and 15), likely because these common word combinations have high likelihood.

**Niche examples.** Perplexity filters can reject niche topics that are otherwise informative, well-written, and contain useful *tail knowledge* of the world. Example 17 contains detailed information about a Manchester art installation but is assigned a high perplexity, likely because it contains uncommon (but valid) word combinations. Examples 20-22 display similar behavior for other niche topics.

### 3.2  DENSITY SAMPLING

**Intuition.** Our intuition is that the data distribution provides a strong coverage signal. High-probability regions contain "prototypical" examples—ones with many near-duplicates and strong representation in the dataset. Low-probability regions will contain outliers, noise, and unique/rare

inputs. If we wish to maximize topic coverage, we should boost the signal from under-represented portions of the input domain and downsample redundant, high-density information.

The key difficulty for our DENSITY sampler is to accurately estimate an example's local density. Like Tirumala et al. (2023) (D4), we assume access to embeddings from a pre-trained LLM. However, we depart from the traditional approach of clustering and opt to sample based on kernel sums. Given a dataset $D$ of embeddings and a kernel $k(x, y)$, we estimate the density using the following score.

$$\text{score}(y) = \sum_{x \in D} k_\lambda(x, y).$$

$\lambda$ is a smoothing parameter called the *kernel bandwidth* that controls the scale of the points' effects. To reduce the complexity from $O(N^2)$ to $O(N \log N)$, we use recent breakthroughs from the algorithm community to approximate the sum (Siminelakis et al., 2019; Coleman & Shrivastava, 2020). Our method resembles that of Coleman et al. (2022), except that (i) we adopt a two-pass sampling algorithm with stronger theoretical guarantees (Theorem C.2) and (ii) we perform the density estimation in the latent space of the model, rather than using Jaccard distances on $n$-grams.

## 3.3 SAMPLING TECHNIQUES

DENSITY and ASK-LLM are both *scoring* methods that reduce an example to a floating point value that measures coverage or quality. Once we have these scores for a complete dataset of training samples, we consider two ways to select examples.

In our experiments, the DENSITY sampler uses IPS to maximize the coverage of the dataset.[1] For our ASK-LLM filter, we adopt top-$k$ sampling because we expect the "yes" probability to be a reliable and strong measure of quality.

## 3.4 RELATIONSHIPS BETWEEN METHODS

**DENSITY, Perplexity, and Loss.** When a language model is trained to minimize perplexity, the LLM itself is a data distribution model. Therefore, the perplexity and loss filtering approaches of Marion et al. (2023), Muennighoff et al. (2023), and other authors can be viewed as model-based density sampling. However, our sampler measures the density of the training dataset in a latent geometric space, while perplexity measures the likelihood under the scoring model. The samplers also differ in terms of decision complexity. Thanks to the capacity of the LLM, a perplexity filter can make highly-nuanced decisions between two texts on the same topic. On the other hand, our DENSITY sampler is constructed from a simple nonparametric density model (Rosenblatt, 1956) that does not have the capacity to distinguish examples at such a granular level.

**ASK-LLM and Perplexity.** Perplexity filters exhibit a strong in-distribution bias, making decisions based on the data used to train the scoring model (not the dataset we wish to sample). By using the LLM for quality evaluation rather than likelihood estimation, our sampler can escape this bias because the additional context and alternative task change the sampling distribution. This occurs even when the ASK-LLM and perplexity models are the same size.

**DENSITY and Clustering.** The kernel sum procedure at the core of DENSITY operates on embedding-similarity relationships in a similar way to SemDeDup and SSL prototypes. Indeed, near-duplicate detection can be viewed as a discretized version of similarity-based density estimation (Kirsch & Mitzenmacher, 2006). Outlier rejection, which motivates the "nearest-to-centroid" heuristic of SSL prototypes, also has intimate connections with density estimation (Schubert et al., 2014).

**Intuition.** Perplexity should be viewed as a "difficulty" or "quality" score rather than as a coverage-maximizing score. Our ASK-LLM sampler should be viewed as a contextualized quality score that incorporates reasoning.[2] Our DENSITY sampler is a pure "coverage" score in the latent representation space, while SemDeDup, and SSL Prototypes all incorporate quality / outlier filtering to some extent (*e.g.*, by preferring points near / far from a centroid).

---

[1] We also implemented top-$K$ and bottom-$K$ sampling, but these samplers do not maintain coverage and perform poorly.

[2] Note that ASK-LLM may also incidentally improve coverage because it does not suffer from in-distribution bias.

## 4 EMPIRICAL SETUP

**Models.** We pre-train T5-style models (Raffel et al., 2020), which belong to the encoder-decoder family of Transformer models and offer competitive performance on many tasks (Shen et al., 2023). See Phuong & Hutter (2022) for a formal introduction to various Transformer model configurations. We train T5-Small (60M) and T5-Large (800M), reusing all of the training settings from the original T5 implementation except the batch size (2048 $\rightarrow$ 1024). We train on batches of 1024 sequences of length 512 for 1M steps. All our experiments are conducted on the TPUv5e architecture.

For perplexity filtering, we experiment with five different language models: T5-{Small, Base, Large, XL, XXL}. For ASK-LLM's quality-scoring model, we consider the FLAN-T5 (Longpre et al., 2023a) models of the same five sizes (Small to XXL), as well as the instruction-tuned Gemma-7B model (Team et al., 2024) (named G.7B for brevity).

**Datasets.** We use the C4 dataset (Raffel et al., 2019) available under the ODC-By license, which was also used for pre-training the original T5 family of models. The C4 dataset is a version of the Common Crawl—a publicly available archive of web-text—that has been pre-processed using several heuristics (Raffel et al., 2020, Section 2.2). In its entirety, C4 contains 184B tokens. We use our algorithms (see Section D for a list) to sample $\{10, 20, 40, 60, 80\}\%$ of C4.

Because a low sampling ratio yields exceedingly small datasets, we choose to train in the iso-compute setting, *i.e.*, training all models for exactly 524B tokens. This results in more epochs (repetitions) at smaller sampling rates. We believe this gives each data curation method an equal chance to maximize model performance, and not penalize methods that sample a small number of high-quality repeatable tokens *vs.* large number of non-repeatable tokens. See Section D, Figure 8 for a demonstration of this process.

**Evaluation.** We use 111 downstream evaluation tasks to assess diverse performance indicators for pre-trained LLMs. On a high-level, we conduct post-finetuning evaluation on GLUE and SuperGLUE for natural language understanding capabilities, as well as benchmark on various knowledge, reasoning, and Q/A evaluation sets after finetuning on the FLANv2 dataset. See Section F for a complete list and further details.

In addition to these individual tasks, to compare a singular *normalized average performance improvement* over all downstream evaluations, we devise a metric called *"Effective Model Size."* It is challenging to concisely summarize performance using a single measure, primarily because our evaluation consists of 111 individual tasks, all of which respond at different rates to data and model optimizations.

Inspired by the LLM scaling-law literature (Hoffmann et al., 2022; Muennighoff et al., 2023), the *"Effective Model Size"* metric measures the effective model size by extrapolating on a parametric fit of the "number of parameters *vs.* downstream eval" trend, averaged over various downstream tasks. See Section E for a formal definition, as well as the fitted scaling laws for the original T5-models on the downstream tasks used in this paper. To summarize, this metric gives a principled answer to the question, *"If using a technique leads to $x$ performance, what size LLM achieves the same $x$ performance if the technique is not used?"*

## 5 EXPERIMENTS

**Does reasoning improve efficiency?** Figure 4 shows that ASK-LLM trains 800M models to an equivalent performance, as if we were to train 1.5B models on the original C4 dataset. ASK-LLM consistently outperforms perplexity filtering (and coverage-maximizing baselines), despite having access to a scoring model of the same model capacity (XL). Similar findings hold for training efficiency (Figure 5). ASK-LLM converges faster than perplexity filters, both in average (expected final performance over all proxy model sizes) and pointwise for the best configuration (Small and XL for training T5-Small and T5-Large).

Figure 7 further demonstrates that prompting adds critical information to the sampler not present in perplexity: ASK-LLM scores show *no correlation* with the perplexity scores. Based on this clear behavioral difference, we conclude that reasoning and context are crucial ingredients. We

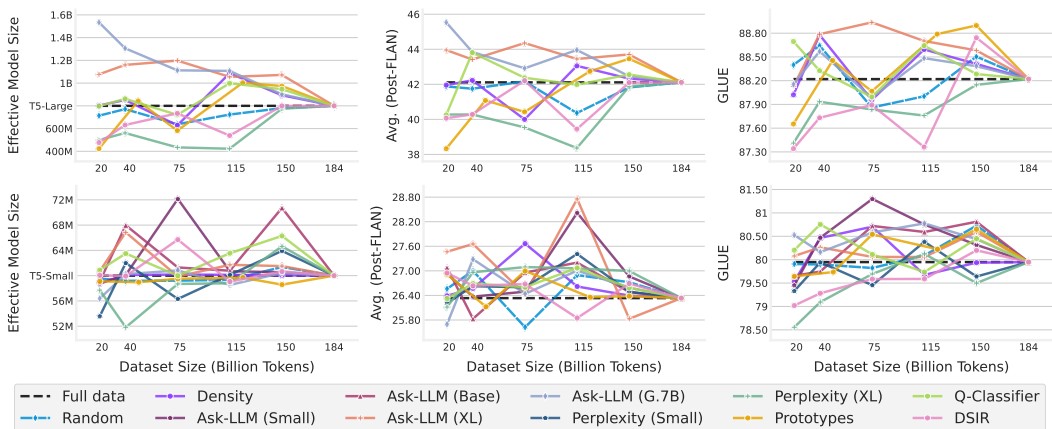

Figure 4: Tradeoff between data quantity (number of unique tokens in the sampled dataset) and model quality for (top) T5-Large and (bottom) T5-Small pre-training. Each point corresponds to a converged pre-training run over a sub-sample. To avoid clutter, not all sampling methods or evaluation metrics are shown in Figure 4 or Table 1; see Section G for the results of all 22 samplers and metrics.

Table 1: Comparison of sampling algorithms at a fixed sample size. For each sampling strategy, we sample the dataset to X% of the original size and pre-train T5-Large for 524B tokens. This table is a cross-section of Figure 4 but with more metrics.

| LLM | Training config. | | Effective Model Size | Downstream tasks | | | | FLAN Instruction Tuning | | | |
| | Sampler | # Tokens | | GLUE | Super GLUE | CNN/ DM | SQuAD | MMLU | BBH | Reasoning | QA |
|---|---|---|---|---|---|---|---|---|---|---|---|
| T5-Large | — | 184B | 800M | 88.2 | 82.5 | 20.8 | 86.7 | 40.7 | 33.6 | 21.6 | 73.0 |
| T5-Large | Random | 18B | 713M | 88.4 | 82.3 | 20.8 | 85.9 | 41.8 | 33.6 | 20.6 | 71.5 |
| T5-Large | Density | 18B | 802M | 88.0 | 80.5 | **20.9** | 86.9 | 42.6 | 35.5 | 19.1 | 70.6 |
| T5-Large | Prototypes | 18B | 423M | 87.7 | 80.5 | 20.4 | 86.6 | 36.7 | 33.0 | 17.6 | 66.0 |
| T5-Large | Perplexity (Small) | 18B | 301M | 87.6 | 80.2 | 20.5 | 85.2 | 36.8 | 33.8 | 17.7 | 60.9 |
| T5-Large | DSIR | 18B | 476M | 87.3 | 81.7 | 20.7 | 85.4 | 39.8 | 33.3 | 22.2 | 65.0 |
| T5-Large | Q-Classifier | 18B | 797M | **88.7** | **83.6** | 20.8 | 87.7 | 40.5 | 35.0 | 20.2 | 65.4 |
| T5-Large | Ask-LLM (G.7B) | 18B | **1.5B** | 88.2 | 82.5 | 20.8 | **87.8** | **44.2** | **37.1** | **22.7** | **78.2** |

expect prompting techniques such as chain-of-thought reasoning (Wei et al., 2022) to further drive performance.

**When are expensive scores justified?** Observing the effective model size while training T5-Large, Figure 4 suggests that other samplers start performing well only at larger sample ratios ($\geq 60\%$), with performance very close to ASK-LLM. On the other hand, at smaller sampling ratios, ASK-LLM tends to significantly outperform both coverage-based samplers, as well as cheaper alternatives for data-quality scoring like Q-Classifier and DSIR (Section D). Hence, the higher costs of LLM-based filters are most justified in two scenarios: (i) improving full-data performance, where quality filtering by removing the lowest-quality data is the main way to push the upper limit of model performance; or (ii) in the low-data regime, where keeping only the highest-quality data drives the most model performance compared to other sampling strategies.

To provide concrete numbers, we report the accelerator-hour costs of training and scoring in Table 2. As described in Section 4, our experiments were run in the iso-compute setting where the goal is to maximize performance at a fixed number of training tokens. To estimate the cost savings, we instead must look at the amount by which we could reduce the training cost while maintaining neutral performance. As an example cost-benefit analysis, consider a setup where AskLLM-T5-XL scores are used to train T5-L. Figure 1 suggests that, conservatively, a 44% reduction in the training horizon is possible without performance loss. This corresponds to a cost of $56\% \times 24 + 10 \approx 23.44$

Table 2: Cost of training and scoring on the C4 dataset. We report the average of 30 training runs.

| Metric (Accelerator-Hr) | T5-XXL | T5-XL | T5-Large | T5-Base | T5-Small |
|---|---|---|---|---|---|
| Scoring Cost (C4) | 49.0 | 10.0 | 1.7 | 0.76 | 0.24 |
| Training Cost (C4) | — | — | 24.0 | — | 9.3 |

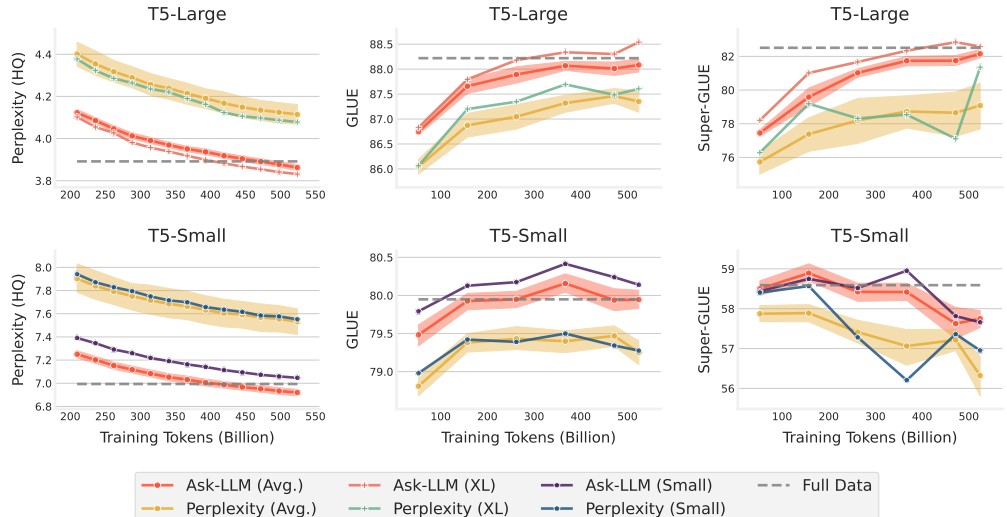

Figure 5: Training efficiency comparison between ASK-LLM and Perplexity filtering, shown by comparing performance of intermediate checkpoints. The (Avg) represents performance *averaged* across (i) scoring model sizes, *i.e.*, T5-{Small, Base, Large, XL, XXL}; and (ii) sampling ratios, *i.e.*, {10, 20, 40, 60, 80}%. The (Small) and (XL) series show the T5-{Small, XL} runs, averaged only over the sampling ratios.

accelerator-hours for AskLLM-sampled training compared to a cost of 24 accelerator-hours without AskLLM, with the benefit that the AskLLM scores can be reused by future runs.

We also observe that random sampling is a strong baseline, aligning with recent observations in the literature. Guo et al. (2022a) found that only three methods outperformed random sampling in a computer vision benchmark of 15 algorithms, and Ayed & Hayou (2023a) prove the existence of adversarial problem instances random sampling is optimal. These results higlight the significance of ASK-LLM's gains.

**Effect of scoring model capacity:** Figure 6 demonstrates a clear scaling trend for ASK-LLM's quality-scoring model: larger scoring models are increasingly beneficial as the scale of the to-be-trained LLM increases. Perplexity filters do not seem to exhibit such trends. The strongly consistent trend suggests that ASK-LLM performance will improve with stronger quality-scoring models – whether obtained by fine-tuning, chain-of-thought prompting, or size. However, we still observe compelling performance even when training large models on data chosen by small ASK-LLM models. For example, ASK-LLM (Base) outperforms perplexity filtering with *any* scoring-model (including T5-XXL) for most sampling ratios (Section F).

**Do samplers select different examples?** We computed the Kendall Tau rank correlation between samplers on 500k examples (Figure 7), finding significant and interesting differences. For example, the "T5-Large" rows show that (i) T5-Large outputs perplexity scores similar to T5-Small early in training, but becomes progressively more nuanced on the path from 20k to 700k training steps, and (ii) perplexity, density, and ASK-LLM select for wildly different criteria, with almost no ranking correlation. This supports our hypothesis that DENSITY prioritizes coverage, representing the original training objective better than any other method besides uniform sampling ( Section G); perplexity re-weights the training objective to up-weight data regions that are in-distribution for the proxy model; and ASK-LLM up-weights the regions that are identified as "high-quality" by the prompt.

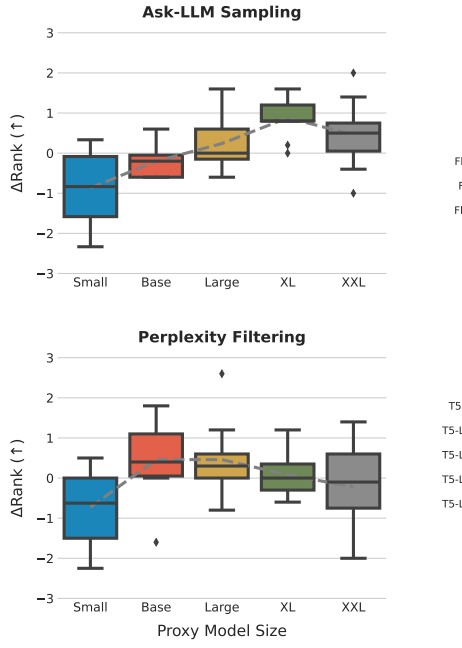

Figure 6: Change in *ranking* of scoring models. A positive ΔRank indicates that the scorer's task-averaged rank increased when training T5-large *vs.* T5-Small.

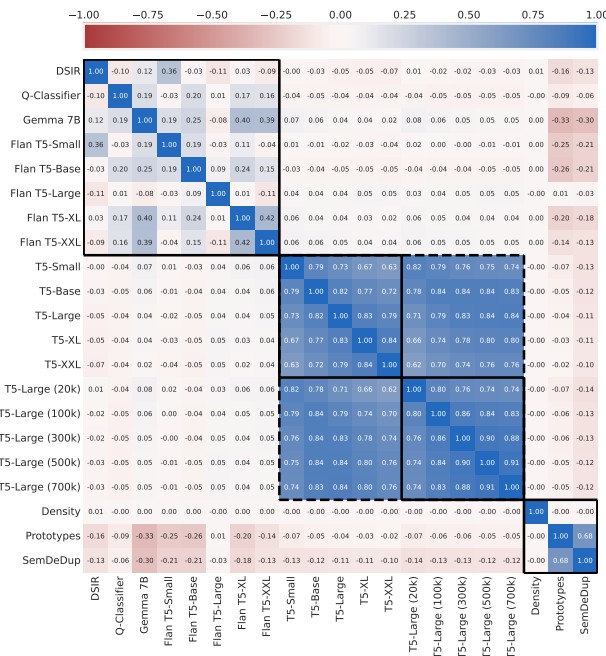

Figure 7: Kendall's Tau correlation amongst the scores from quality filters (first 8), perplexity filters (next 10), and coverage-based samplers (last 3).

## 6 DISCUSSION

**Amortized scoring.** The ASK-LLM and perplexity scorers require considerable computation—one LLM inference call for every training sample—which is concerning from both a carbon-emissions and cost perspective (Strubell et al., 2019). However, we argue that the scoring costs are *amortized over many pre-training runs*, which together cost significantly more than the ASK-LLM inference calls (Luccioni et al., 2023). In practical systems, cheaper samplers / scoring models can also pre-filter examples for our more expensive scorers. While LLM pre-training is often thought of as a one-time cost, this has historically not been the case. We therefore view quality scores as a long-term investment. See Section C.1 for a deeper discussion about the cost of ASK-LLM scoring.

**LLM-Based Data Refinement.** Recursively training on model-generated data causes degradation in both diffusion models and LLMs, inciting concerns about whether the internet will remain a viable source of training data (Shumailov et al., 2023; Alemohammad et al., 2023; Briesch et al., 2023). The primary mechanism for this degradation is a mode-collapse behavior where the self-consumption loop amplifies the in-distribution bias of the LLM. It is therefore somewhat surprising that LLMs are so effective at deciding which training data to consume.

We rigorously examine the bias characteristics of ASK-LLM in Figure 11 (Appendix G), finding a much greater variance in topic affinity among the T5 models when the score is obtained via ASK-LLM versus perplexity. Therefore, we conclude that the ASK-LLM biases depend more strongly on the fine-tuning / prompting of the model than on in-distribution bias (since all models in the T5 family share the same training distribution bias). These results raise important questions about whether LLM-based filters can function as an intervention in the self-consumption loop, allowing LLMs to self-improve.

**LLM and Prompt Sensitivity.** While we are unable to present rigorous ablations of the prompt, we have observed poor calibration with numeric score outputs and Likert rubrics (unless highly-detailed, case-by-case grading criteria are provided). This results in noisy labels for borderline data and degenerate distributions that make sampling difficult. One known method to improve the calibration

of LLM raters is to transform scalar-output tasks into token-probability tasks (Ren et al., 2023), which may explain why we observe relatively robust performance with our $\Pr[\text{Yes}]$ scores. We do examine the sensitivity of scoring decisions to the choice of LLM judge, with detailed comparisons in Appendix H. The general rule is that larger model sizes can distinguish quality in more nuanced ways (e.g., advertisement copy is selected by small judges but not by large ones).

One can cheaply guess whether a new prompt or LLM might yield different downstream training performance by measuring the rank correlation between scores: high correlation ($> 0.8$) is a sufficient condition for similar performance. However, the converse is not true – two judges can achieve similar results even if they disagree on many points. We hypothesize that this occurs because the performance of a model trained on sampled data is an aggregate property of the training set. In practice, we observe that the different ASK-LLM judges are not strongly correlated on individual score pairs ( Figure 7), even though they attain similar downstream performance.

**Decoder-Only Models.** The experiments in this work address the data-efficiency of encoder-decoder models in the T5 family. The encoder-decoder structure is still the subject of research and industrial use, especially for applications that can employ asymmetric encoder and decoder components to exploit efficiency tradeoffs (Zhang et al., 2025). However, the vast majority of today's research and practical applications focus on decoder-only models.

While we do not conduct experiments with decoder-only architectures in this paper, we would like to note that a variant of our ASK-LLM technique was used in the development of the Gemma 3 family of models. Specifically, the ASK-LLM framework was employed to re-weight the pretraining corpus to reduce occurrences of low-quality data (Kamath et al., 2025).

## 7 CONCLUSION

We studied the performance of sampling algorithms that select high-quality data through highly-capable proxies and maximize coverage through embedding similarity. Our experiments reveal that LLM-based quality filtering yields a Parteo optimal efficiency tradeoff between data quantity and model quality, with important implications for training cost, self-improvement, and LLM training data curation.

## ACKNOWLEDGEMENTS

We sincerely thank Xinyun Chen and Kelvin Guu for their insightful feedback on early drafts of this paper. We would also like to thank Sammy Jerome, Anton Tsitsulin, Peilin Zhong, and Marc Brockschmidt for their insights and technical discussions as we refined the ASK-LLM technique. We thank Zhang Li for his significant engineering contributions to the project. We also thank the Gemma 3 team for being our first production users – especially Alek Andreev, who helped create the opportunity, and Cassidy Hardin, who worked tirelessly to productionize ASK-LLM. Finally, we wish to thank Fernando Pereira, Vahab Mirrokni, Armand Joulin, Andrew Dai, and Ya Xu for their strong leadership and support for this project.

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

# Appendices

## A   IMPACT STATEMENT

While increased LLM accessibility has well-documented risks, we expect data-efficient pre-training to be a net social good that reduces (amortized) carbon emissions and pre-training cost while improving quality.

## B   LIMITATIONS

While the paper pushes the frontier of LLM-training from both quality and efficiency fronts by better curating pre-training datasets, we would also like to note the limitations of this paper that both better informs the reader and hopefully guides future work. First, due to the sheer cost of training LLMs even once, let alone doing data-efficiency research, we only train one kind of transformer models (encoder-decoder) and only on the C4 dataset. The transferability of our results to more popular, decoder-only models and larger datasets is still yet to be explored, and an interesting direction for future work. Next, due to T5-models' inability to code (primarily a tokenization issue), we don't include any coding evaluations in this paper. Curating high-quality coding data is an interesting and active direction of research (Gunasekar et al., 2023). Further, all of our evaluations are limited to post-finetuning (as is the prevalent setting with T5 models), hence the effect of our data-curation techniques on zero/few-shot prompting is also not clear.

## C   ALGORITHMS

### C.1   ASK-LLM SAMPLING

---

**Algorithm 1** ASK-LLM Sampling

---

**Input:** Dataset $\mathcal{D} = \{x_1, x_2, \cdots, x_N\}$ s.t. $x_i \in \mathcal{X}$ is the training sample in plain-text, sample size $k$, scoring model $\mathcal{M} : \mathcal{X}; \mathcal{X} \mapsto \mathbb{R}$
**Output:** Sampled data

1:   Initialize list of scores $S = []$.
2:   **for** $n = 1 \rightarrow N$ **do**
3:      $\text{prompt}_n \leftarrow \text{make\_prompt}(x_n)$             ▷ Make ASK-LLM prompts as in Figure 3
4:      Append $\mathcal{M}(\text{"yes"} \mid \text{prompt}_n)$ to $S$                ▷ Use $\mathcal{M}$ to score $x_n$
5:   **return** $k$ elements from $\mathcal{D}$ with top-$k$ scores in $S$, without replacement.

---

**Discussion on the cost of ASK-LLM scoring.** Even though ASK-LLM sampling results in impressive performance and training efficiency improvements compared to training on the full-dataset (Section G), the data quality scoring cost might seem prohibitive. On the other hand, on top of the improved results, we argue the following to be compelling points in justifying ASK-LLM's one-time-amortized data scoring cost:

- ASK-LLM only requires *forward passes* on the entire dataset. This is much cheaper than (i) training the model itself which requires both forward and backward passes on multiple repetitions of the entire dataset, (ii) gradient-based data-curation techniques (Sachdeva & McAuley, 2023; Sachdeva et al., 2023) that also require backward passes, *etc.*

- An additional benefit of the ASK-LLM framework is the ability to leverage memory-efficient, quantized LLM inference setups (Dettmers et al., 2022). This is strictly not possible, *e.g.*, for pre-training LLMs. Notably, quantization isn't the only ASK-LLM-friendly technique. All the recent (and future) advances in efficient *inference* techniques for LLMs (Weng, 2023) directly reduce the amortization cost of the ASK-LLM framework.

- Another benefit of ASK-LLM is the ability to naïvely parallelize quality scoring. To be more specific, we can simply scale-up the amount of *small & independent* inference resources, and run inference calls for various training samples parally. Note that inference hardware has much smaller requirements compared to, *e.g.*, pre-training or fine-tuning requirements. This is primarily true because of no batch size requirement for inference *vs.* large batch size requirement while training. This enables scaling-up hardware to happen via a large number of small-compute setups (*e.g.*, 4 interconnected GPUs per node) versus increasing the number of large-compute setups (*e.g.*, 1000s of interconnected GPUs per node).

- ASK-LLM also uses strictly less compute compared to teacher-student knowledge distillation based training setups (Agarwal et al., 2023). This is true simply because knowledge distillation require (i) bigger teacher model's softmax predictions (ii) for each token in our training data. On the other hand, ASK-LLM requires just the score of the token "yes" given the prompt.

## C.2   DENSITY SAMPLING

Our density sampler is adapted from that of Coleman et al. (2022), with a few critical departures:

- We use a two-pass procedure that allows for more rigorous theoretical guarantees (and different sampling behavior).
- We conduct the density estimation in the model's latent space rather than using Jaccard similarity over $n$-grams.

**Improvements:** Jaccard similarities are sufficient to construct a reasonable sampling distribution for genomics applications, which are significantly more structured than natural language. However, this is not the case with text — we found that sampling based on Jaccard density is no better than random. For this reason, we must use different kernels ($p$-stable rather than MinHash) and different input representations (embedding rather than $n$-grams).

However, our more interesting departure from Coleman et al. (2022) is our two-pass sampling procedure, which changes the behavior of the algorithm and allows for more rigorous theoretical guarantees. The original method was only able to demonstrate convergence of cluster populations in the sampled dataset. While this leads to (weak) convergence for some measures of diversity, it also requires strong assumptions about the cluster structure.

**Theory:** We use a recent result that demonstrates consistent sketch-based estimation of the kernel sum (Theorem 3.3 of Liu et al. (2023)), which we paraphrase below.

**Lemma C.1.** *Let $P(x)$ denote a probability density function. Let $\mathcal{D} \underset{\text{iid}}{\sim} P(x)$ denote a dataset. Let $k(x, y)$ be a positive definite LSH kernel, and let $S$ be the DENSITY score. Then $S(x)$ is a consistent estimator for the kernel sum.*

$$S(x) \underset{\text{i.p.}}{\to} \frac{1}{N} \sum_{x_i \in \mathcal{D}} k(x_i, q)$$

*with convergence rate $O(\sqrt{\log R/R})$.*

If we perform inverse propensity sampling using the score in Theorem C.1, we obtain a sampling procedure that outputs a uniformly-distributed sample.

**Theorem C.2.** *Let $Q(x)$ be the distribution formed by (i) drawing $N$ samples i.i.d. from a distribution $P$, e.g. $\mathcal{D} = \{x_1, ...x_N\} \sim P$, and (ii) keeping $x$ with probability proportional to $\frac{1}{S(x)}$. Under the conditions of Lemma C.1, $Q(x) \underset{\text{i.p.}}{\to} U(x)$, where $U(x)$ is the uniform distribution.*

*Proof.* Under the conditions of Wied & Weißbach (2012) (specifically, positive-definiteness and $\ell_1$ integrability / bounded domain), the kernel sum is a consistent estimator of the density. That is, the sum converges in probability to $P(x)$.

$$\frac{1}{N} \sum_{x_i \in \mathcal{D}} k(x_i, q) \underset{\text{i.p.}}{\to} P(x)$$

Theorem C.1 shows that $S(x)$ converges in probability to the sum (and thus to $P(x)$). By Slutsky's Theorem, $\frac{1}{S(x)} \to \frac{1}{P(x)}$ for all $x$ in the support of the distribution (i.e. $P(x) \neq 0$). The probability of generating $x$ as part of the sample is:

$$Q(x) = \Pr[\text{Select} x \cap \text{Generate} x] = \Pr[\text{Select} x]\Pr[\text{Generate} x] = \frac{1}{S(x)}P(x)$$

Because $\frac{1}{S(x)} \to \frac{c}{P(x)}$ for some constant $c$, we have that $Q(x) \to c$. □

Theorem C.2 demonstrates that our DENSITY sampler outputs a uniformly-distributed collection of points over the input space (latent LLM representation space).

---

**Algorithm 2** Inverse Propensity Sampling (IPS) via Kernel Density Estimation (KDE)

---

**Input:** Dataset $\mathcal{D} = \{x_1, x_2, \cdots, x_N\}$ of embeddings, sample size $k$, kernel $k$ with corresponding locality-sensitive hash family $\mathcal{H}$ (see Coleman & Shrivastava (2020)), hash range $B$, rows $R$, random seed $s$
**Output:** Sampled data

1: **Initialize:** KDE sketch $\mathcal{S} \leftarrow 0^{R \times B}$
2: Generate $R$ independent hash functions $h_1, \ldots, h_R$ from $\mathcal{H}$ with range $B$ and random seed $s$.
3: **for** $n = 1 \to N$ **do** ▷ Construct KDE estimator for $D$.
4:     **for** $r = 1 \to R$ **do** ▷ Add $x_n$ to the KDE estimator.
5:         $\mathcal{S}_{r,h_r(x_n)} + = 1$
6: Initialize list of scores $S = []$.
7: **for** $n = 1 \to N$ **do** ▷ Score each example $x_n$
8:     score $= 0$
9:     **for** $r = 1 \to R$ **do** ▷ Compute approximate KDE using $\mathcal{S}$
10:         score$+ = \mathcal{S}[r, h_r(x_n)]$
11:     Append score$/R$ to $S$
12: **return** $k$ elements from $\mathcal{D}$ with probability $p = \frac{\sum S}{S}$ without replacement.

---

**Cost:** Like SemDeDup, D4, and SSL prototypes, our DENSITY sampler requires access to embeddings for each example in the training corpus. However, by eliminating the expensive clustering step, we eliminate a significant computational overhead. Our DENSITY sampling routine required just 80MB of memory and two linear passes through the dataset to score all 364M embeddings. This is significantly less expensive than clustering.

**Tuning:** We also eliminate a large number of hyperparameters, improving tuning. Cluster-based samplers must choose the number of clusters, clustering optimizer and objective, and per-cluster sampling rate or deduplication similarity. Kernel density estimation, on the other hand, has just *two* hyperparameters: the choice of kernel and the bandwidth. We did not observe a significant performance variation among different bandwidth and kernel choices (e.g., the L2 and cosine kernels of Coleman & Shrivastava (2020) perform nearly identically). This is likely because all positive-definite kernels enjoy strong guarantees on the distribution approximation error (Devroye, 1983).

## D    DATA-CURATION TECHNIQUES

### D.1    RANDOM SAMPLING

The de-facto standard for obtaining samples of large datasets where we sample training examples uniformly at random. Notably, random sampling has also been accompanied with strong results in a variety of applications in the data-curation literature primarily due to its unbiased sampling (Ayed & Hayou, 2023b; Guo et al., 2022b).

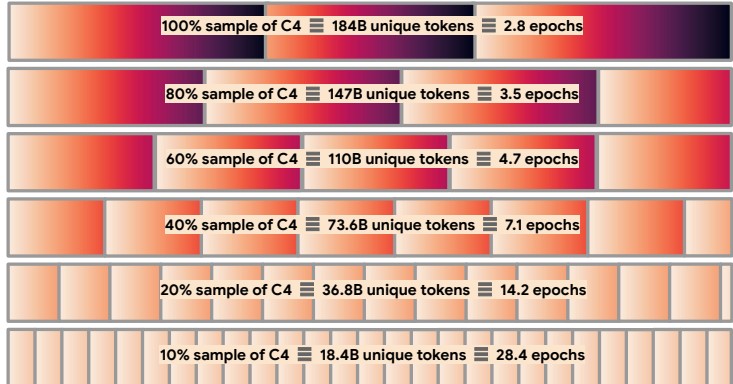

Figure 8: We consider a setup where all of our models are trained on exactly 524B tokens, causing us to repeat the same examples for more epochs when we downsample. We borrow the format of this graphic from Muennighoff et al. (2023), who consider a similar setting.

## D.2 DENSITY SAMPLING

See Section 3.2 for technical details about the DENSITY sampler. We use Sentence-T5-Base (Ni et al., 2021) as our embedding model for training samples, primarily due to its contrastive training, giving confidence for computing distances amongst its $768$-dim embeddings. We use the PStable hash (Datar et al., 2004) to hash the embeddings, along with a $[1,000 \times 20,000]$ sketch matrix.

## D.3 SEMDEDUP

The key idea is to perform (coverage maximizing) semantic deduplication inside clusters of the original dataset (**?**). We re-use the Sentence-T5-Base embeddings of data-points (Section D.2), and perform $k$-means clustering to obtain $10,000$ clusters of the entire dataset.

## D.4 SSL PROTOTYPES

They key idea is to remove *prototypical* points in a dataset (Sorscher et al., 2022). As a meaningful proxy, this method removes the points closest to cluster centroids of a dataset. For brevity, we use the name "Prototypes" when reporting our results. We re-use the same embeddings and clustering for both SemDeDup and Prototypes.

## D.5 PERPLEXITY FILTERING

A popular quality-filtering approach in the literature is to use the perplexity of proxy language models to filter data-points with a high-perplexity under that language model. While the literature historically used small language models for perplexity filtering (Wenzek et al., 2019; Muennighoff et al., 2023), recent work (Marion et al., 2023) suggests improved filtering performance when using LLMs for this task. To this end, we employ perplexity filtering with T5-{Small, Base, Large, XL, XXL} models; as well as intermediate checkpoints during the course of training T5-Large: {20k, 100k, 300k, 500k, 700k}.

## D.6 TEXT-QUALITY CLASSIFIER (Q-CLASSIFIER)

First proposed by GPT-3 for curating its pretraining dataset (Brown et al., 2020), and later used by various state-of-the-art LLMs at their time (Chowdhery et al., 2023; Anil et al., 2023; Gao et al., 2020; Du et al., 2022) another popular quality filtering approach is to train a linear classifier for distinguishing web-scrape data *vs.* known reference high-quality data. Consistent with existing usage of this technique (Gao et al., 2020; Xie et al., 2024; Brown et al., 2020; Chowdhery et al., 2023; Du et al., 2022), we train a hashing-based linear classifier with a hash size of 262k trained to classify if a document is either from (negative) C4 or (positive) Wikipedia + BookCorpus. We train this classifier

for a total of 218k steps (equivalent to 14 Trillion unigrams), and based on recent evidence (Xie et al., 2024) we sample the documents with the highest score according to this classifier.

### D.7 DATA SELECTION WITH IMPORTANCE RESAMPLING (DSIR)

Proposed by Xie et al. (2024), DSIR performs importance sampling using a bag-of-words estimator (we use unigram and bigram features) over some "high-quality" target data-source. This approach is, in spirit, quite similar to the aforementioned text-quality classification approach but performs distribution-matching in a non-parametric way, and without the hassle of training a classifier on large piles of data. To be consistent, we use Wikipedia + BookCorpus as the target source for DSIR as well. We re-use the official public implementation for DSIR[3].

### D.8 ASK-LLM SAMPLING

See Section 3.1 for technical details about the ASK-LLM sampler. Since ASK-LLM relies on the reasoning capabilities of instruction-tuned models, we use the Flan-T5-{Small, Base, Large, XL, XXL} (Longpre et al., 2023a) and instruction tuned Gemma-7B (Team et al., 2024) models for obtaining the quality scores in ASK-LLM.

## E EFFECTIVE MODEL SIZE

It is challenging to concisely summarize performance using a single measure, primarily because our evaluation consists of 111 individual tasks, all of which respond at different rates to data and model optimizations. To provide a holistic view of performance, we fit a parametric model, *a.k.a* scaling law (Hoffmann et al., 2022; Muennighoff et al., 2023), of the "model-size ↔ quality" curve for the original T5 models (*i.e.*, trained on the full C4 dataset) over various downstream tasks.

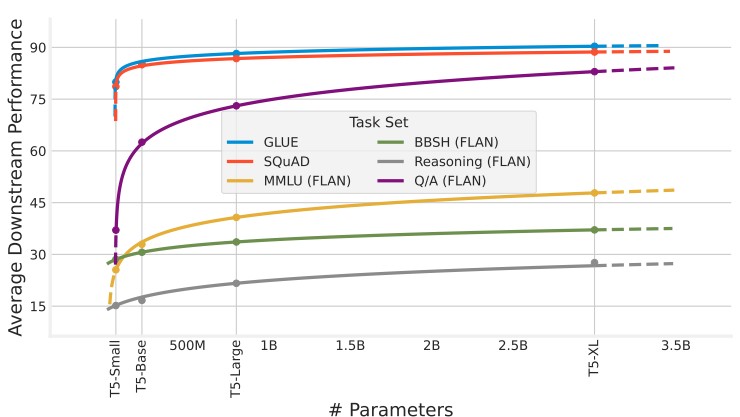

Figure 9: Empirical scaling laws for T5-models trained on the entire C4 dataset for various downstream tasks.

More specifically, we fit functions of the following form, for each downstream task separately:

$$\text{ModelSize} = A + \exp(B + \text{EvalPerformance} * C) \,, \qquad (1)$$

and use `scipy.optimize.curve_fit` to estimate the $A, B, C$ parameters based on the evaluations of T5-{Small, Base, Large, XL}. See Figure 9 for a visual interpretation of the parametric models we fit.

Finally, given the performance of a model trained on downsampled data, the **effective model size** is defined as the predicted model size by plugging in the observed downstream performance into the parametric scaling law estimated via Equation (1), taking a median over various downstream evaluation tasks listed in Figure 9.

---

[3]https://github.com/p-lambda/dsir

# F    DOWNSTREAM EVALUATION TASKS

## F.1    PERPLEXITY

Defined as the exponentiated average negative log-likelihood of an average sequence in the dataset; we report the perplexity computed over the target tokens in T5's denoising objective (Raffel et al., 2020) over the default validation set provided by C4. Note that C4's validation set is a random sample of the dataset, so it is prone to be of much lower quality than curated sources, and hence, a less reliable indicator of true model quality.

## F.2    HQ PERPLEXITY

As our best effort to devise an inexpensive-to-compute metric that is better aligned with model quality than perplexity on C4's validation set, inspired by the evaluation conducted in Tirumala et al. (2023), we construct a *high-quality* validation set from non web-scrape sources. We collate the validation sets from (1) English portion of wiki40b (Guo et al., 2020), (2) realnews and webtext subsets of C4, and (3) news commentary from the LM1B dataset (Chelba et al., 2013).

## F.3    GLUE

A popular natural language understanding meta-benchmark comprising of eleven different tasks (Wang et al., 2018). Note that we report the average score for all individual tasks, after finetuning on the concatenation of all individual tasks' training sets, as is done in the original T5 implementation.

## F.4    SUPERGLUE

A harder meta-benchmark (*vs.* GLUE) built to further test the natural language understanding abilities of language models (Wang et al., 2019). Similar to GLUE, we report the average score of all tasks, and conduct fine-tuning on all tasks' concatenated train-set.

## F.5    CNN/DM

We use the CNN/DM dataset (Hermann et al., 2015) for testing our models' abstractive summarization abilities. Like the T5 original setting, we finetune on the train-set, and report the ROUGE-2 scores.

## F.6    SQUAD

A popular dataset (Rajpurkar et al., 2016) used to evaluate question-answering capabilities of language models, we compare the finetuned performance of our models using exact-match as the metric.

## F.7    FLAN INSTRUCTION TUNING

A popular application of LLMs has been instruction-following, and chatting capabilities. To test our model's quality on this front, we finetune our models on the FLANv2 dataset (Longpre et al., 2023a), and test the instruction-tuned models' performance from four fronts:

- 5-shot MMLU (Hendrycks et al., 2020): a popular benchmark consiting of exam questions from 57 tasks.
- 3-shot Big Bench Hard (BBH) (Srivastava et al., 2022): a popular set of 23 hardest tasks from big bench.
- Reasoning: macro-average 8-shot performance on GSM8k (Cobbe et al., 2021), SVAMP (Patel et al., 2021), ASDIV (Miao et al., 2021), and StrategyQA (Geva et al., 2021) benchmarks.
- QA: macro-average 0-shot performance on UnifiedQA (Khashabi et al., 2020), BoolQ (Clark et al., 2019), Arc-Easy and Arc-Challenge (Clark et al., 2018) benchmarks.
- Average: macro-average of all the four benchmarking suites listed above: MMLU, BBH, Reasoning, and Q/A.

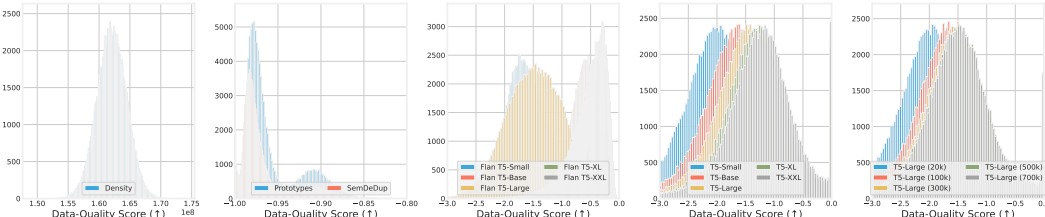

Figure 10: Score distribution of various data curation techniques. The plots for Flan-T5-* models are for ASK-LLM, whereas ones using T5-* models are for perplexity filtering.

Please note that all of our reported numbers are based on *single checkpoint* evaluations, *i.e.*, we first select the best checkpoint during FLAN finetuning using the *average* performance on all tasks, and report the individual task performance on that checkpoint.

## G    ADDITIONAL RESULTS

### G.1    (FIGURE 10) QUALITY-SCORE DISTRIBUTION FOR DIFFERENT SAMPLERS

For different data curation techniques listed in Section D, we examine the distribution of estimated *data-quality* scores normalized in a way that higher represents better data quality.

- For the DENSITY sampler, the plotted score is proportional to the likelihood of the example under the kernel density estimate.

- For the Prototypes sampler, the plotted score represents the negated cosine similarity of data-point with its assigned cluster centroid.

- For the SemDeDup sampler, the plotted score represents the negated maximum cosine similarity of a datapoint to all other datapoints in its respective cluster.

- For the perplexity filtering sampler, the plotted score represents the negated perplexity of a training sample.

- For the ASK-LLM sampler, the plotted score represents the log probability of the token "`yes`" given the prompt in Figure 3.

### G.2    (FIGURE 11) ANALYSIS FOR DIFFERENT SAMPLERS' AFFINITY TO DIFFERENT TOPICS

Since the different sampling strategies explored in this paper operate with different implicit biases, we try to understand if certain samplers exhibit more affinity to certain topics in the data compared to others. To visualize this phenomenon, we conduct the following procedure:

1. Load a random sample of 500k datapoints, along with their respective data-quality scores.

2. Perform topic-modeling (via LDA) on the 500k datapoints with 9 topics.

3. Manually inspect the most common word associations in each of the 9 topics, and label a "high-level description" for each topic.

4. Assign each of the 500k datapoints to the LDA topic with the highest likelihood and analyze the differences between the distribution of scores within each topic and the global score distribution.

5. We conducted a one-way ANOVA, one-vs-rest style, to determine whether the averages were statistically significant. Because $N = 500k$, all effects were significant at the $p < 0.01$ level. We measure the effect size using Cohen's d and report results in  Figure 11.

From the topic affinity analysis in Figure 11, we can observe a few interesting common trends:

- The perplexity filters have relatively low variance in their scores, indicating a much less biased sampling. This is expected, because perplexity filtering primarily biases toward "well-written text" which is relatively task/topic agnostic.

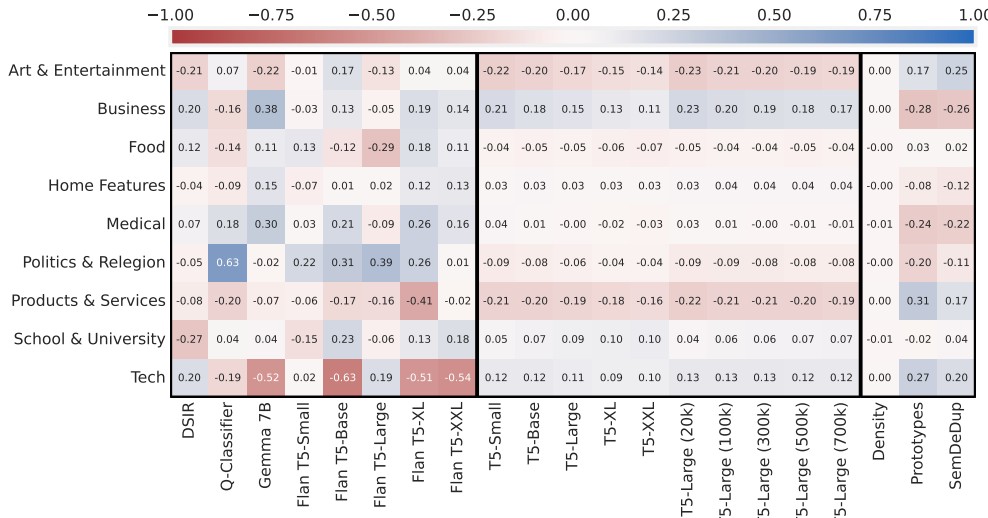

Figure 11: Estimated topic affinity for quality filters (first 8), perplexity filters (next 10), and coverage-based samplers (last 3) over 500k randomly selected training samples. A higher score represents more affinity. All effects significant at the $p < 0.01$ level.

- The quality-based samplers (ASK-LLM, DSIR, Q-Classifier) exhibit a much stronger variance in their scores, with a common liking towards business, political, and religious content; and a common disliking towards tech, art, and entertainment content.

- Consistent with the score correlations in Figure 7, the prototypes and SemDeDup samplers exhibit inverse correlation with most other samplers when comparing topic affinity too.

- Density sampling, as expected, exhibits no special affinity to any particular topic because it's objective is to only maximize coverage.

### G.3 (FIGURES 12 TO 20) DATA-QUANTITY *vs.* MODEL-QUALITY FOR DIFFERENT SAMPLERS

For different data curation techniques listed in Section D, we investigate the tradeoff between the sampling rate and the respectively trained model's quality on various downstream evaluations listed in Section F. We plot our results in the following figures:

- (Figure 12) **T5-Small, coverage**: Pre-training T5-Small on different amounts of data sampled by {Random sampling, DENSITY sampling, Self-supervised Prototypes sampling, SemDeDup}.

- (Figure 13) **T5-Large, coverage**: Pre-training T5-Large on different amounts of data sampled by {Random sampling, DENSITY sampling, Self-supervised Prototypes sampling, SemDeDup}.

- (Figure 14) **T5-Small, ASK-LLM**: Pre-training T5-Small on different amounts of data sampled by ASK-LLM using the {Flan-T5-Small, Flan-T5-Base, Flan-T5-Large, Flan-T5-XL, Flan-T5-XXL} scoring models.

- (Figure 15) **T5-Large, ASK-LLM**: Pre-training T5-Large on different amounts of data sampled by ASK-LLM using the {Flan-T5-Small, Flan-T5-Base, Flan-T5-Large, Flan-T5-XL, Flan-T5-XXL} scoring models.

- (Figure 16) **T5-Small, Other quality-based Filters**: Pre-training T5-Small on different amounts of data sampled by {Random sampling, DSIR, Q-Classifier, ASK-LLM (G.7B), ASK-LLM (XL)} scoring models.

- (Figure 17) **T5-Large, Other quality-based Filters**: Pre-training T5-Large on different amounts of data sampled by {Random sampling, DSIR, Q-Classifier, ASK-LLM (G.7B), ASK-LLM (XL)} scoring models.

- (Figure 18) **T5-Small, Perplexity filtering**: Pre-training T5-Small on different amounts of data sampled by Perplexity filtering using the {T5-Small, T5-Base, T5-Large, T5-XL, T5-XXL} scoring models.

- (Figure 19) **T5-Large, Perplexity filtering**: Pre-training T5-Large on different amounts of data sampled by Perplexity filtering using the {T5-Small, T5-Base, T5-Large, T5-XL, T5-XXL} scoring models.

- (Figure 20) **T5-Large, Perplexity filtering**: Pre-training T5-Large on different amounts of data sampled by Perplexity filtering using the {20k, 100k, 300k, 500k, 700k} intermediate checkpoints of T5-Large as data quality scoring models.

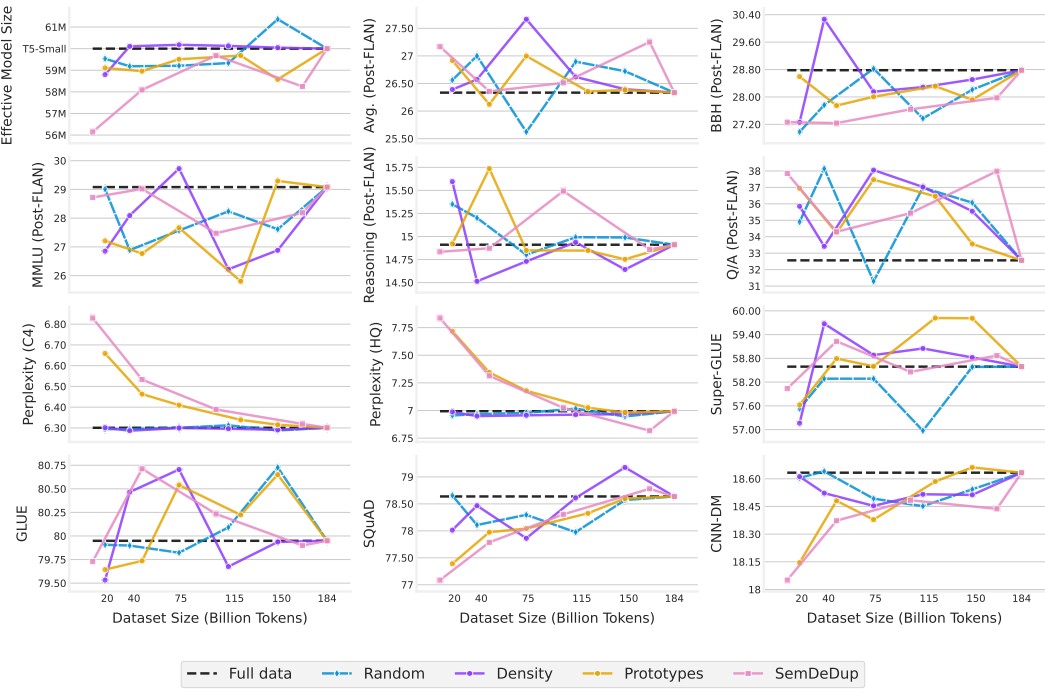

Figure 12: Tradeoff between data quantity and model quality while pre-training T5-Small. Each point in this plot comes from the converged pre-training run over a sampled dataset. See Section F for a description about the metrics used in this plot.

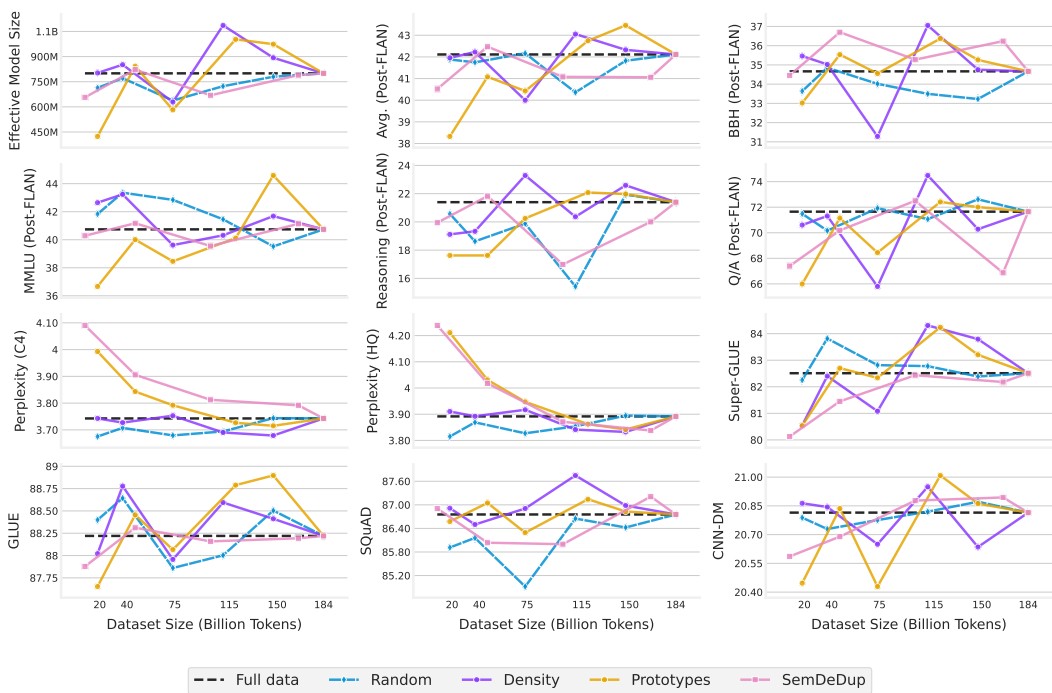

Figure 13: Tradeoff between data quantity and model quality while pre-training T5-Large. Each point in this plot comes from the converged pre-training run over a sampled dataset. See Section F for a description about the metrics used in this plot.

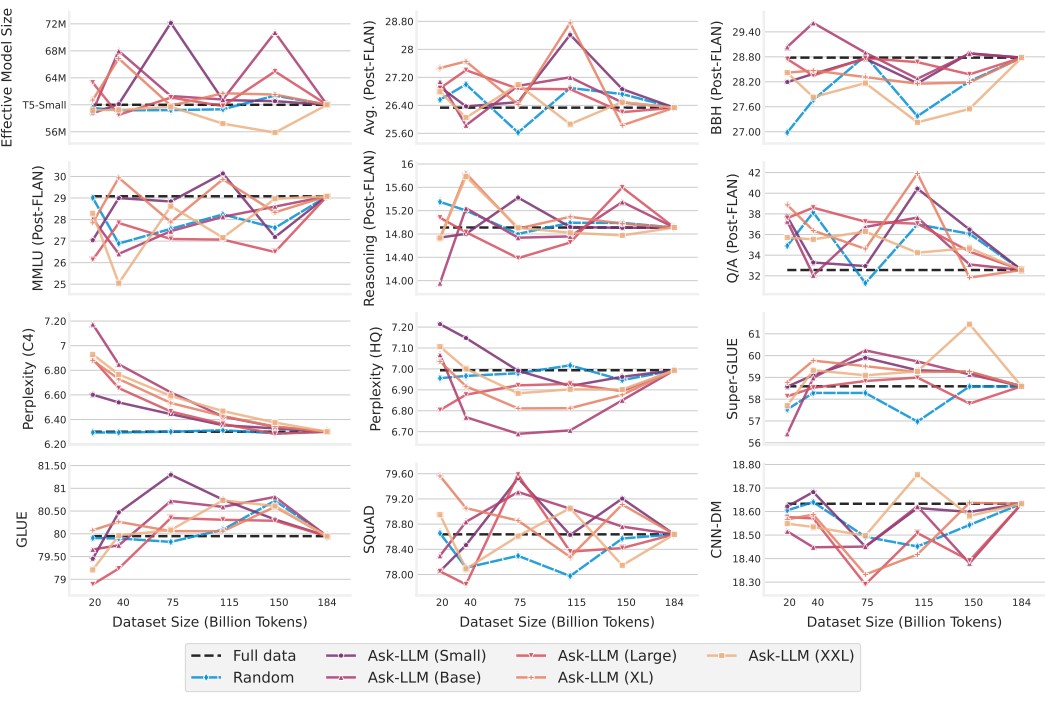

Figure 14: Tradeoff between data quantity and model quality while pre-training T5-Small. Each point in this plot comes from the converged pre-training run over a sampled dataset. See Section F for a description about the metrics used in this plot.

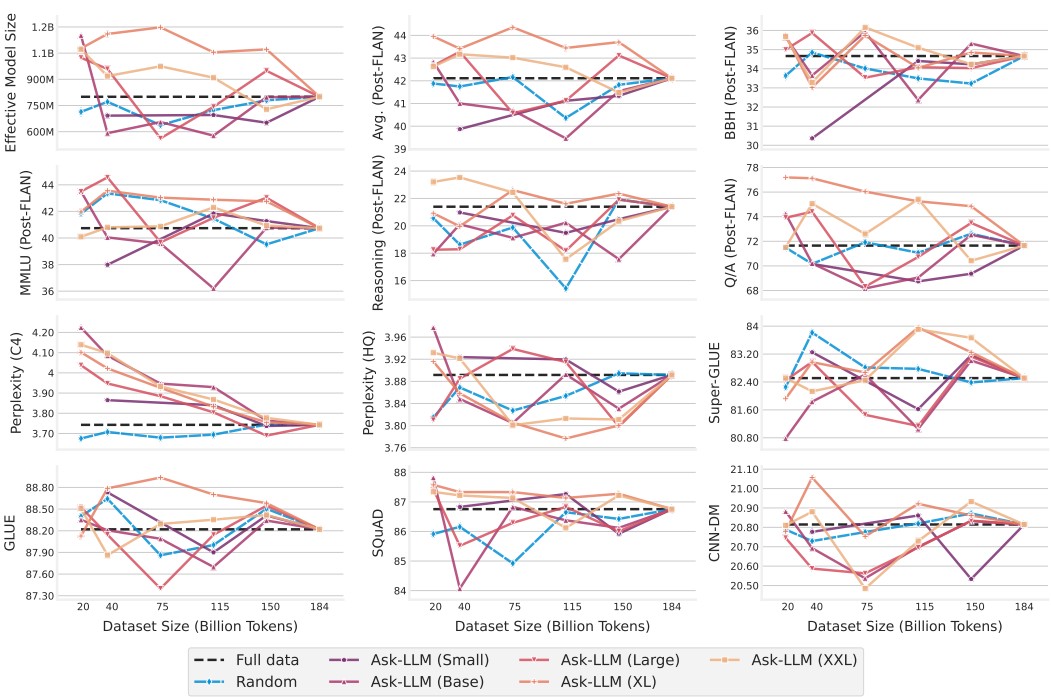

Figure 15: Tradeoff between data quantity and model quality while pre-training T5-Large. Each point in this plot comes from the converged pre-training run over a sampled dataset. See Section F for a description about the metrics used in this plot.

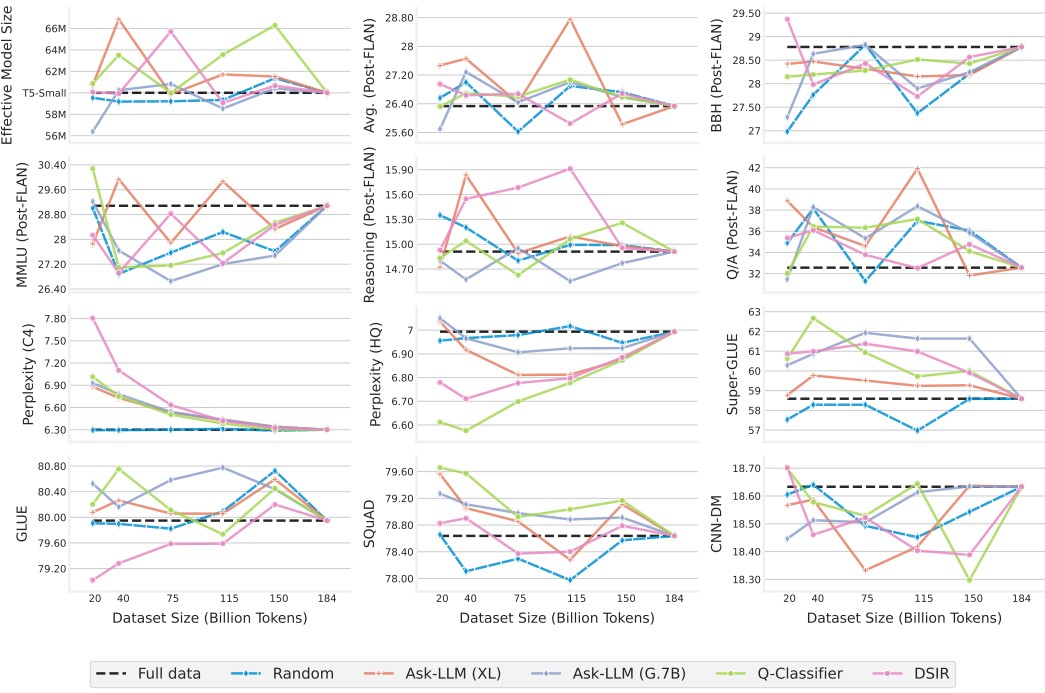

Figure 16: Tradeoff between data quantity and model quality while pre-training T5-Small. Each point in this plot comes from the converged pre-training run over a sampled dataset. See Section F for a description about the metrics used in this plot.

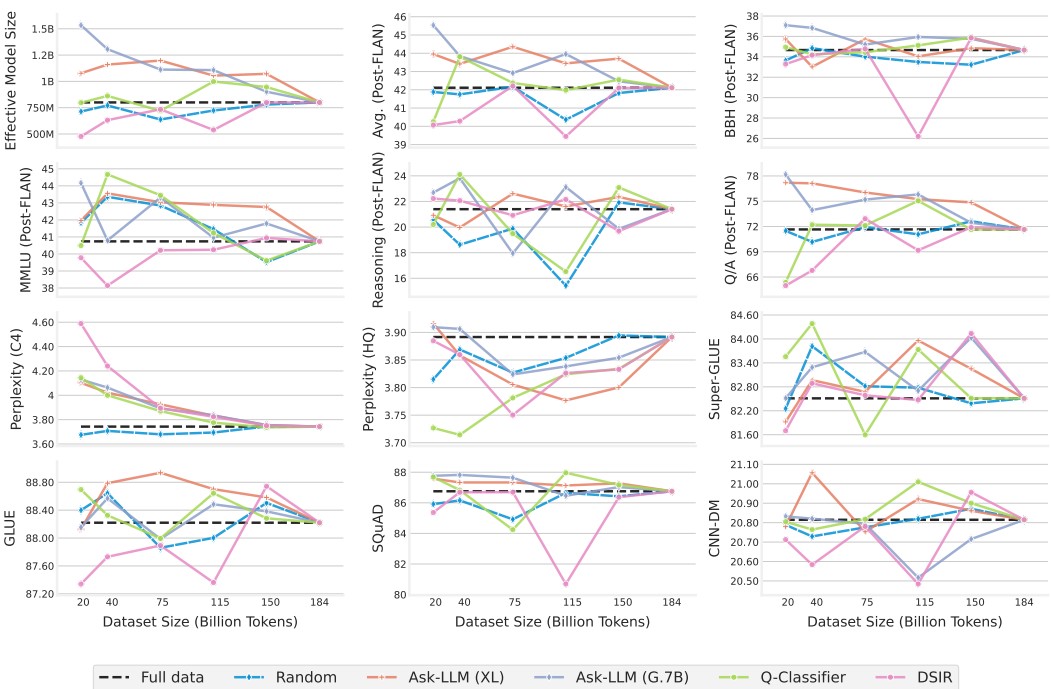

Figure 17: Tradeoff between data quantity and model quality while pre-training T5-Large. Each point in this plot comes from the converged pre-training run over a sampled dataset. See Section F for a description about the metrics used in this plot.

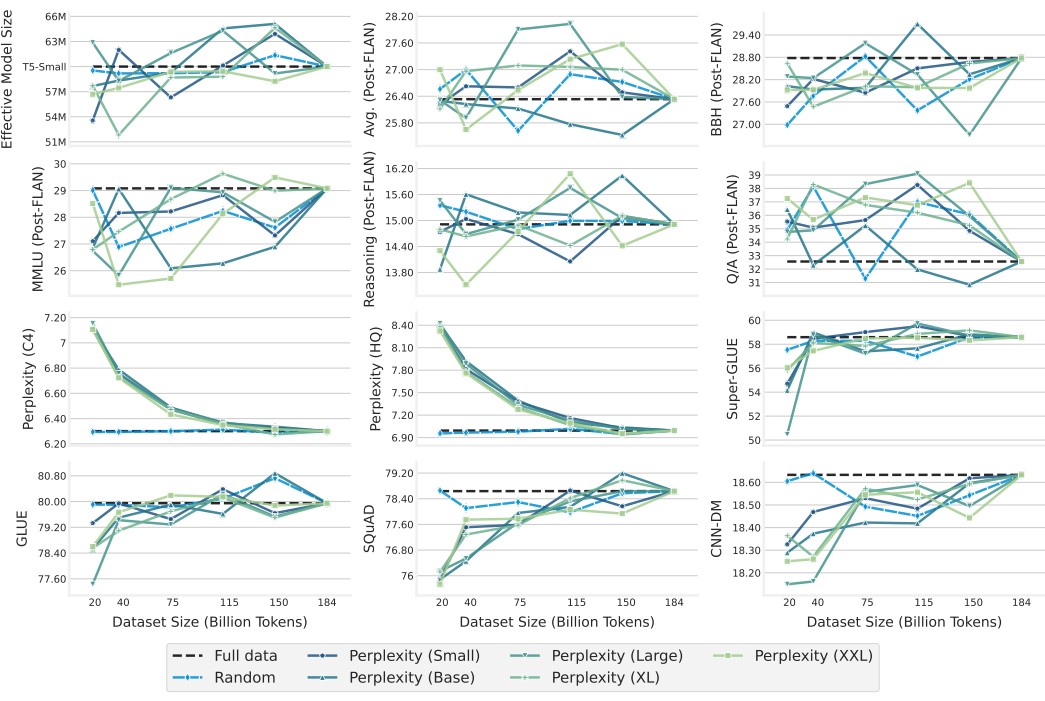

Figure 18: Tradeoff between data quantity and model quality while pre-training T5-Small. Each point in this plot comes from the converged pre-training run over a sampled dataset. See Section F for a description about the metrics used in this plot.

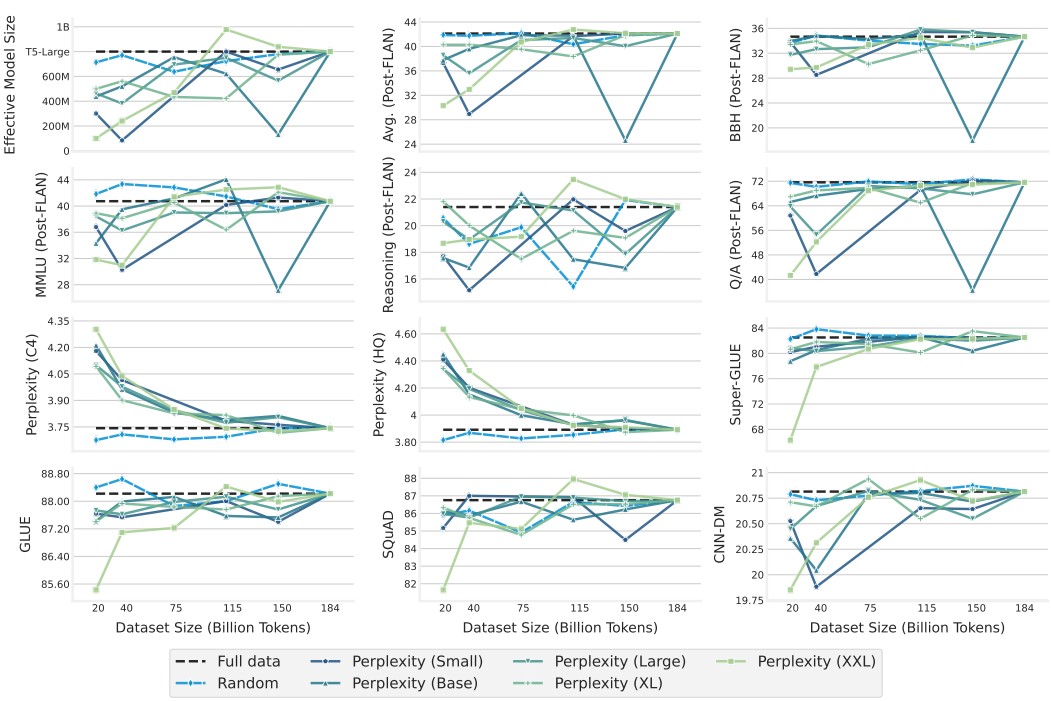

Figure 19: Tradeoff between data quantity and model quality while pre-training T5-Large. Each point in this plot comes from the converged pre-training run over a sampled dataset. See Section F for a description about the metrics used in this plot.

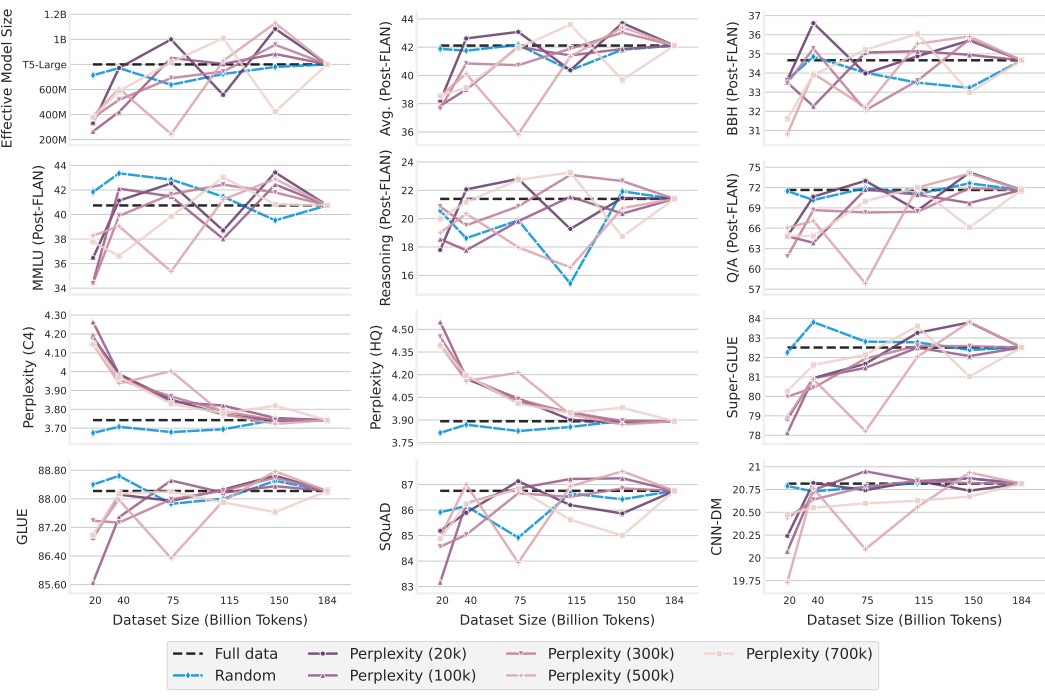

Figure 20: Tradeoff between data quantity and model quality while pre-training T5-Large. Each point in this plot comes from the converged pre-training run over a sampled dataset. See Section F for a description about the metrics used in this plot.

## G.4 (FIGURES 21 TO 29) QUALITY OF FRESH *vs.* REPEATED TOKENS FOR DIFFERENT SAMPLERS

We investigate the data-efficiency for different data curation techniques listed in Section D over various downstream evaluations listed in Section F, when stratifying by the maximum number of repetitions allowed over the sampled dataset. We plot our results in the following figures:

- (Figure 21) **T5-Small, coverage**: Average data-efficiency of pre-training T5-Small on data sampled by {Random sampling, DENSITY sampling, Self-supervised Prototypes sampling, SemDeDup}, stratified by the maxmimum number of allowed repetitions over the sampled dataset.

- (Figure 22) **T5-Large, coverage**: Average data-efficiency of pre-training T5-Large on data sampled by {Random sampling, DENSITY sampling, Self-supervised Prototypes sampling, SemDeDup}, stratified by the maxmimum number of allowed repetitions over the sampled dataset.

- (Figure 23) **T5-Small, ASK-LLM**: Average data-efficiency of pre-training T5-Small on data sampled by ASK-LLM using the {Flan-T5-Small, Flan-T5-Base, Flan-T5-Large, Flan-T5-XL, Flan-T5-XXL} scoring models, stratified by the maxmimum number of allowed repetitions over the sampled dataset.

- (Figure 24) **T5-Large, ASK-LLM**: Average data-efficiency of pre-training T5-Large on data sampled by ASK-LLM using the {Flan-T5-Small, Flan-T5-Base, Flan-T5-Large, Flan-T5-XL, Flan-T5-XXL} scoring models, stratified by the maxmimum number of allowed repetitions over the sampled dataset.

- (Figure 25) **T5-Small, Other quality-based Filters**: Pre-training T5-Small on different amounts of data sampled by {Random sampling, DSIR, Q-Classifier, ASK-LLM (G.7B), ASK-LLM (XL)} scoring models, stratified by the maxmimum number of allowed repetitions over the sampled dataset.

- (Figure 26) **T5-Large, Other quality-based Filters**: Pre-training T5-Large on different amounts of data sampled by {Random sampling, DSIR, Q-Classifier, ASK-LLM (G.7B), ASK-LLM (XL)} scoring models, stratified by the maxmimum number of allowed repetitions over the sampled dataset.

- (Figure 27) **T5-Small, Perplexity filtering**: Average data-efficiency of pre-training T5-Small on data sampled by Perplexity filtering using the {T5-Small, T5-Base, T5-Large, T5-XL, T5-XXL} scoring models, stratified by the maxmimum number of allowed repetitions over the sampled dataset.

- (Figure 28) **T5-Large, Perplexity filtering**: Average data-efficiency of pre-training T5-Large on data sampled by Perplexity filtering using the {T5-Small, T5-Base, T5-Large, T5-XL, T5-XXL} scoring models, stratified by the maxmimum number of allowed repetitions over the sampled dataset.

- (Figure 29) **T5-Large, Perplexity filtering**: Average data-efficiency of pre-training T5-Large on data sampled by Perplexity filtering using the {20k, 100k, 300k, 500k, 700k} intermediate checkpoints of T5-Large as data quality scoring models, stratified by the maxmimum number of allowed repetitions over the sampled dataset.

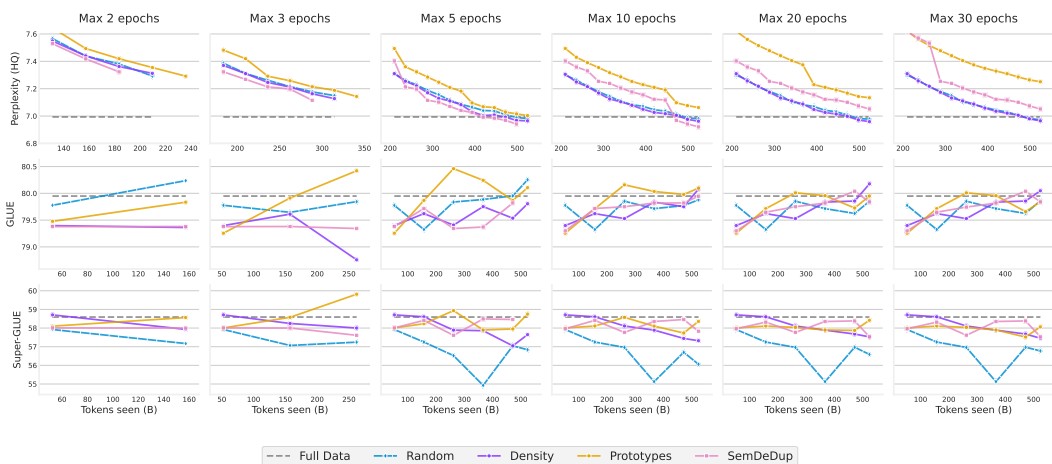

Figure 21: Average data-efficiency of pre-training T5-Small on sampled data, stratified by maximum number of allowed repetitions on the sampled dataset. Each point in this plot represents the performance of an intermediate checkpoint *averaged* over all sampling ratios, as long as the maximum allowed repetitions have not been reached. See Section F for a description about the metrics used in this plot.

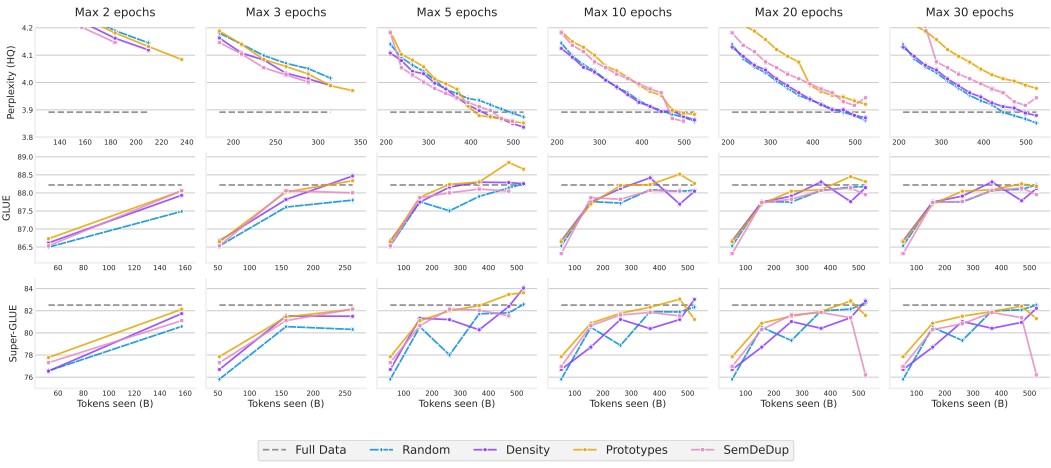

Figure 22: Average data-efficiency of pre-training T5-Large on sampled data, stratified by maximum number of allowed repetitions on the sampled dataset. Each point in this plot represents the performance of an intermediate checkpoint *averaged* over all sampling ratios, as long as the maximum allowed repetitions have not been reached. See Section F for a description about the metrics used in this plot.

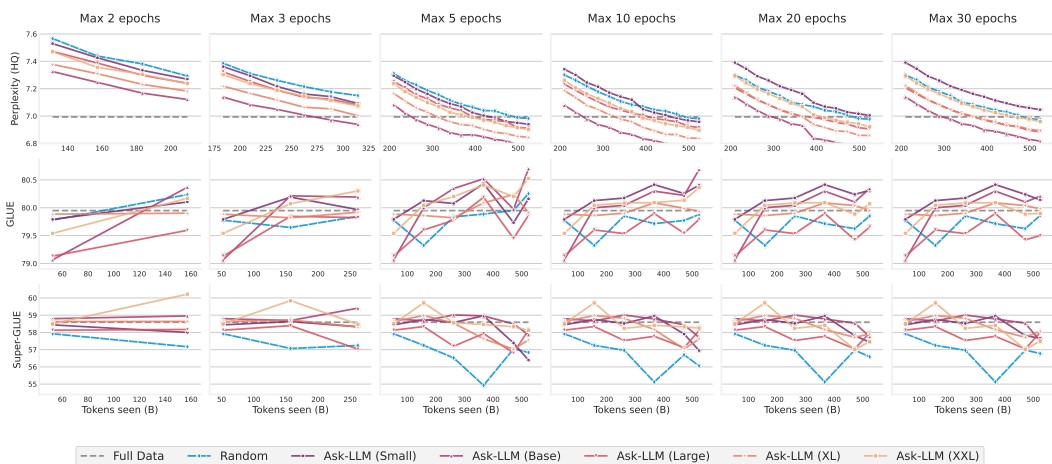

Figure 23: Average data-efficiency of pre-training T5-Small on sampled data, stratified by maximum number of allowed repetitions on the sampled dataset. Each point in this plot represents the performance of an intermediate checkpoint *averaged* over all sampling ratios, as long as the maximum allowed repetitions have not been reached. See Section F for a description about the metrics used in this plot.

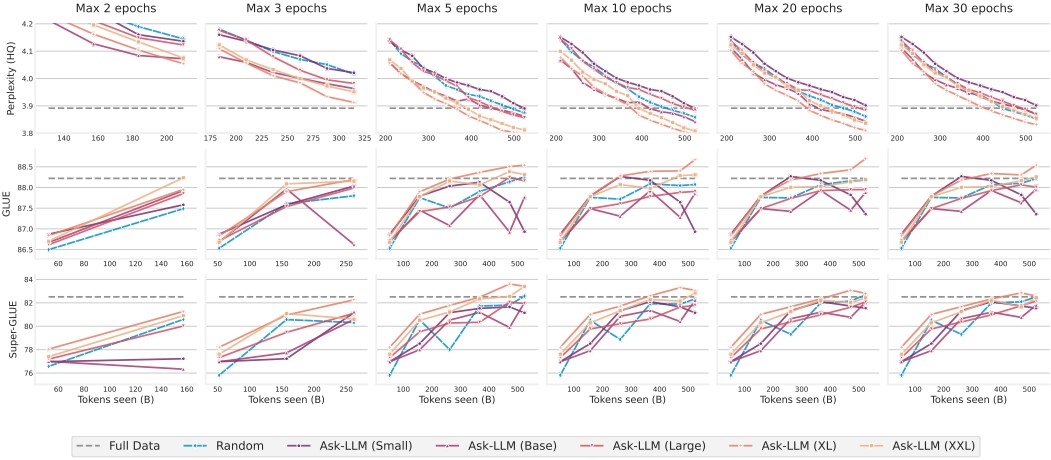

Figure 24: Average data-efficiency of pre-training T5-Large on sampled data, stratified by maximum number of allowed repetitions on the sampled dataset. Each point in this plot represents the performance of an intermediate checkpoint *averaged* over all sampling ratios, as long as the maximum allowed repetitions have not been reached. See Section F for a description about the metrics used in this plot.

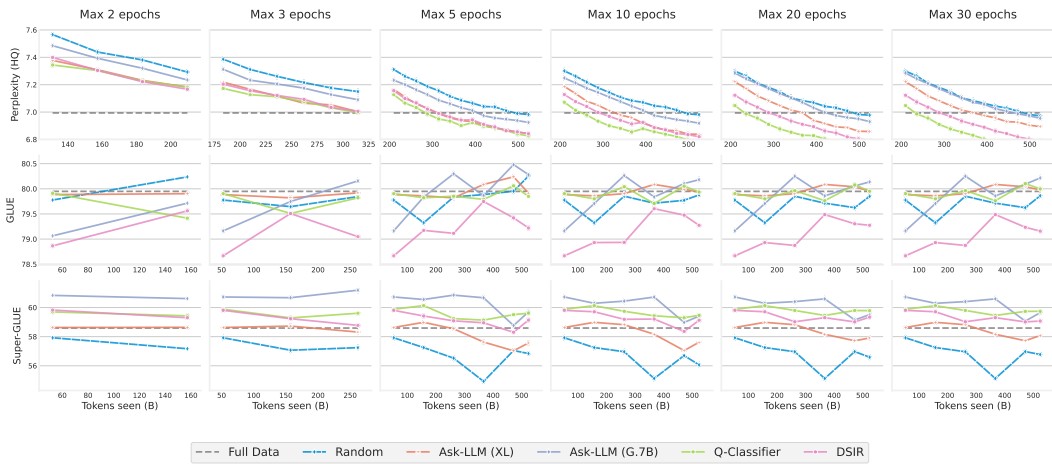

Figure 25: Average data-efficiency of pre-training T5-Small on sampled data, stratified by maximum number of allowed repetitions on the sampled dataset. Each point in this plot represents the performance of an intermediate checkpoint *averaged* over all sampling ratios, as long as the maximum allowed repetitions have not been reached. See Section F for a description about the metrics used in this plot.

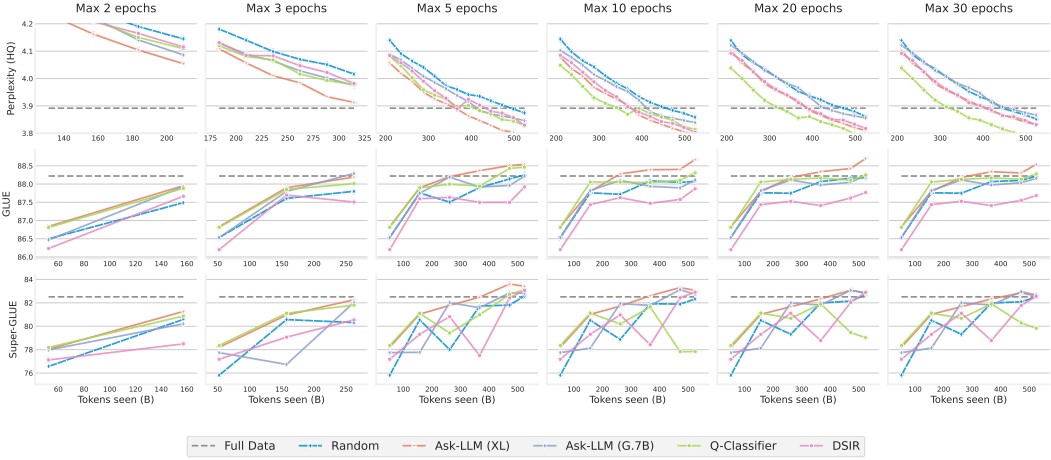

Figure 26: Average data-efficiency of pre-training T5-Large on sampled data, stratified by maximum number of allowed repetitions on the sampled dataset. Each point in this plot represents the performance of an intermediate checkpoint *averaged* over all sampling ratios, as long as the maximum allowed repetitions have not been reached. See Section F for a description about the metrics used in this plot.

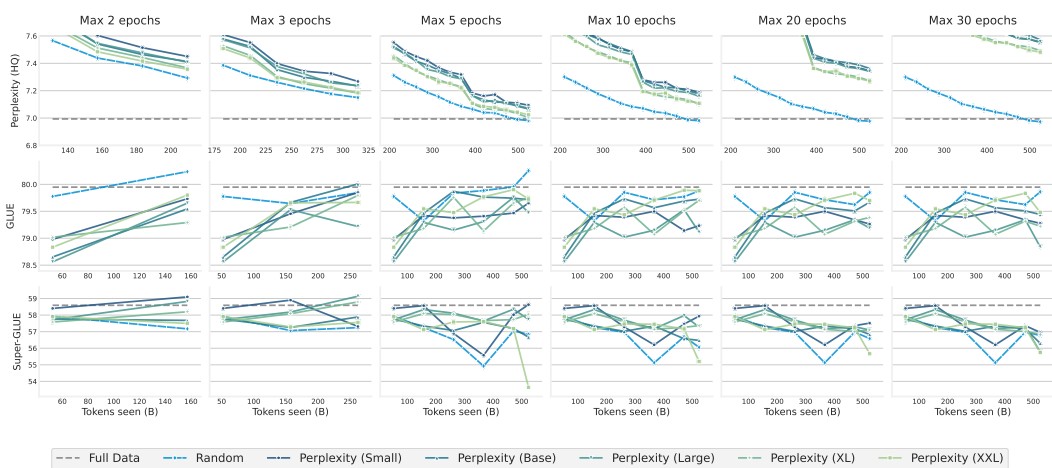

Figure 27: Average data-efficiency of pre-training T5-Small on sampled data, stratified by maximum number of allowed repetitions on the sampled dataset. Each point in this plot represents the performance of an intermediate checkpoint *averaged* over all sampling ratios, as long as the maximum allowed repetitions have not been reached. See Section F for a description about the metrics used in this plot.

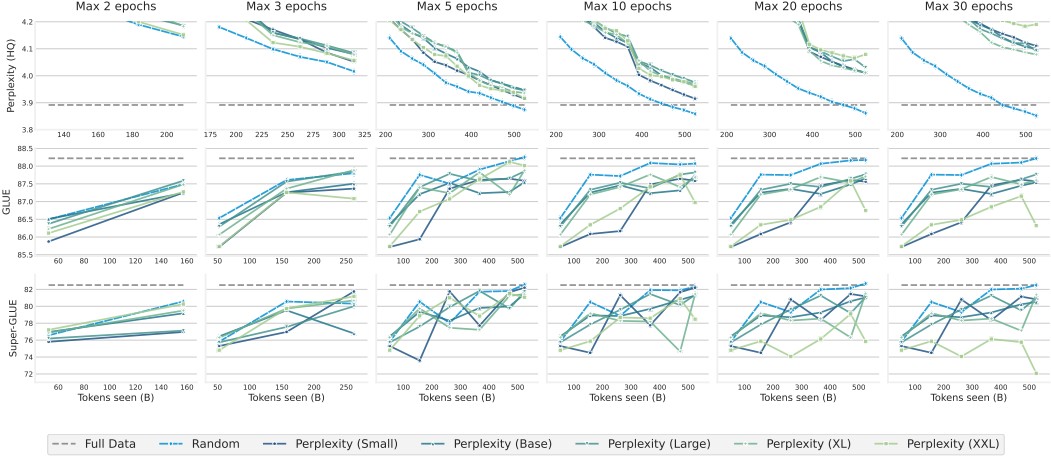

Figure 28: Average data-efficiency of pre-training T5-Large on sampled data, stratified by maximum number of allowed repetitions on the sampled dataset. Each point in this plot represents the performance of an intermediate checkpoint *averaged* over all sampling ratios, as long as the maximum allowed repetitions have not been reached. See Section F for a description about the metrics used in this plot.

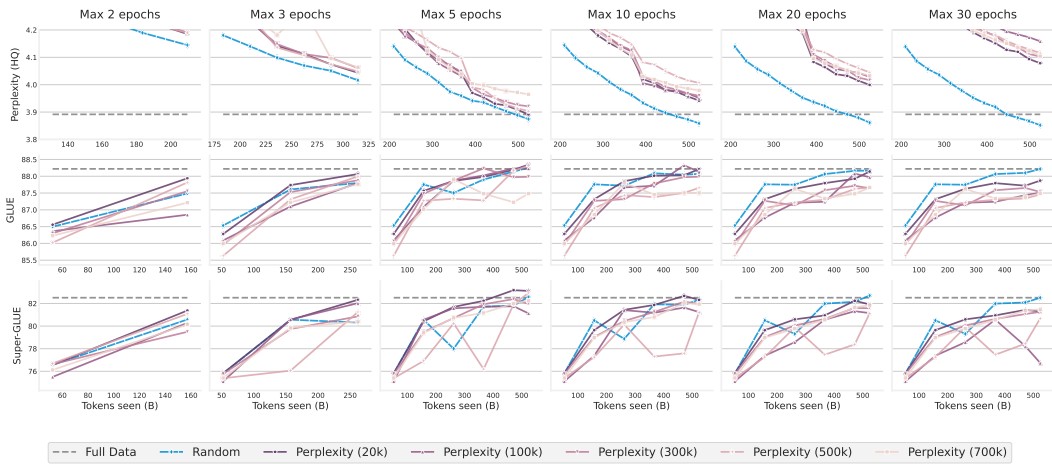

Figure 29: Average data-efficiency of pre-training T5-Large on sampled data, stratified by maximum number of allowed repetitions on the sampled dataset. Each point in this plot represents the performance of an intermediate checkpoint *averaged* over all sampling ratios, as long as the maximum allowed repetitions have not been reached. See Section F for a description about the metrics used in this plot.

## G.5   (FIGURES 30 TO 36) DATA-EFFICIENCY OF DIFFERENT SAMPLERS

We investigate the data-efficiency for different data curation techniques listed in Section D over various downstream evaluations listed in Section F, when stratifying by the sampling ratio *or* the size of the sampled dataset. We plot our results in the following figures:

- (Figure 30) **T5-Small, ASK-LLM**: Data-efficiency of pre-training T5-Small on data sampled by ASK-LLM using the {Flan-T5-Small, Flan-T5-Base, Flan-T5-Large, Flan-T5-XL, Flan-T5-XXL} scoring models, stratified by the sampling ratio.

- (Figure 31) **T5-Large, ASK-LLM**: Data-efficiency of pre-training T5-Large on data sampled by ASK-LLM using the {Flan-T5-Small, Flan-T5-Base, Flan-T5-Large, Flan-T5-XL, Flan-T5-XXL} scoring models, stratified by the sampling ratio.

- (Figure 32) **T5-Small, Other quality-based Filters**: Data-efficiency of pre-training T5-Small on data sampled by {Random sampling, DSIR, Q-Classifier, ASK-LLM (G.7B), ASK-LLM (XL)} scoring models, stratified by the sampling ratio.

- (Figure 33) **T5-Large, Other quality-based Filters**: Data-efficiency of pre-training T5-Large on data sampled by {Random sampling, DSIR, Q-Classifier, ASK-LLM (G.7B), ASK-LLM (XL)} scoring models, stratified by the sampling ratio.

- (Figure 34) **T5-Small, Perplexity filtering**: Data-efficiency of pre-training T5-Small on data sampled by Perplexity filtering using the {T5-Small, T5-Base, T5-Large, T5-XL, T5-XXL} scoring models, stratified by the sampling ratio.

- (Figure 35) **T5-Large, Perplexity filtering**: Data-efficiency of pre-training T5-Large on data sampled by Perplexity filtering using the {T5-Small, T5-Base, T5-Large, T5-XL, T5-XXL} scoring models, stratified by the sampling ratio.

- (Figure 36) **T5-Large, Perplexity filtering**: Data-efficiency of pre-training T5-Large on data sampled by Perplexity filtering using the {20k, 100k, 300k, 500k, 700k} intermediate checkpoints of T5-Large as data quality scoring models, stratified by the sampling ratio.

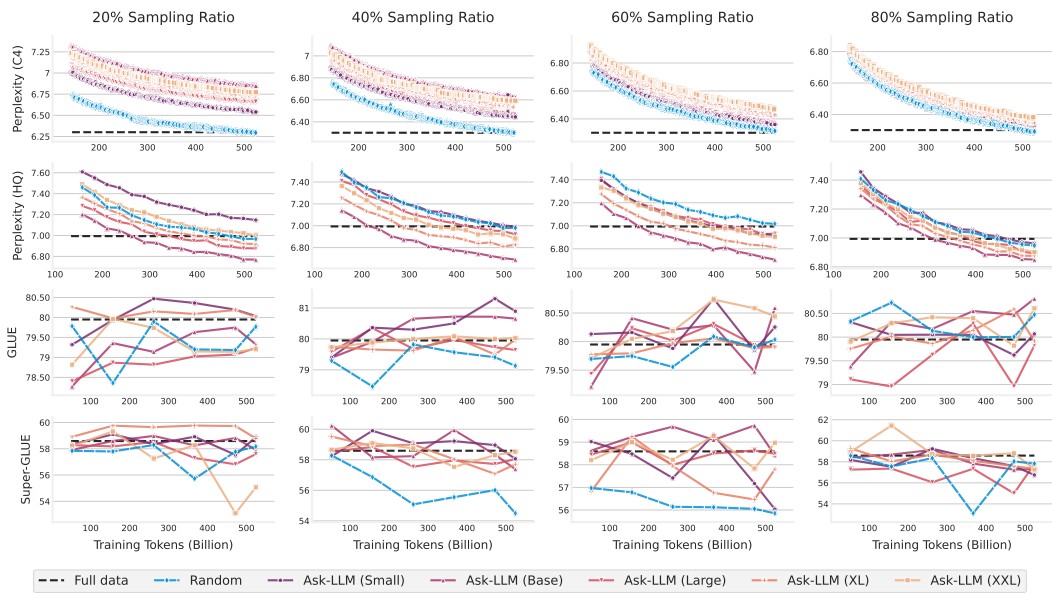

Figure 30: Data efficiency comparison of different samplers while training T5-Small for various sampling ratios. Each point in this plot is the performance of an intermediate checkpoint during the course of training on sampled data.

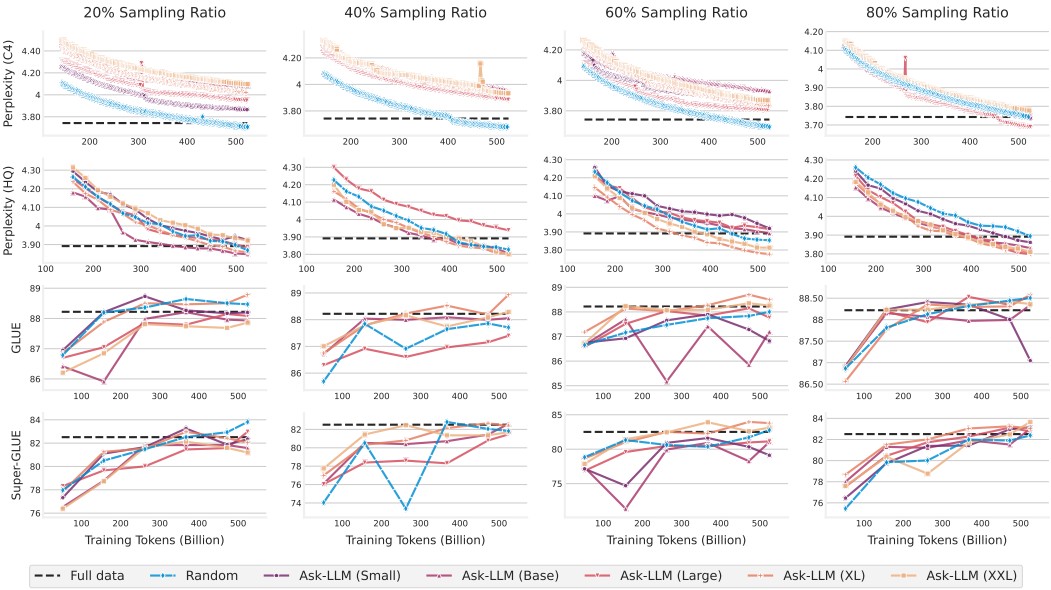

Figure 31: Data efficiency comparison of different samplers while training T5-Large for various sampling ratios. Each point in this plot is the performance of an intermediate checkpoint during the course of training on sampled data.

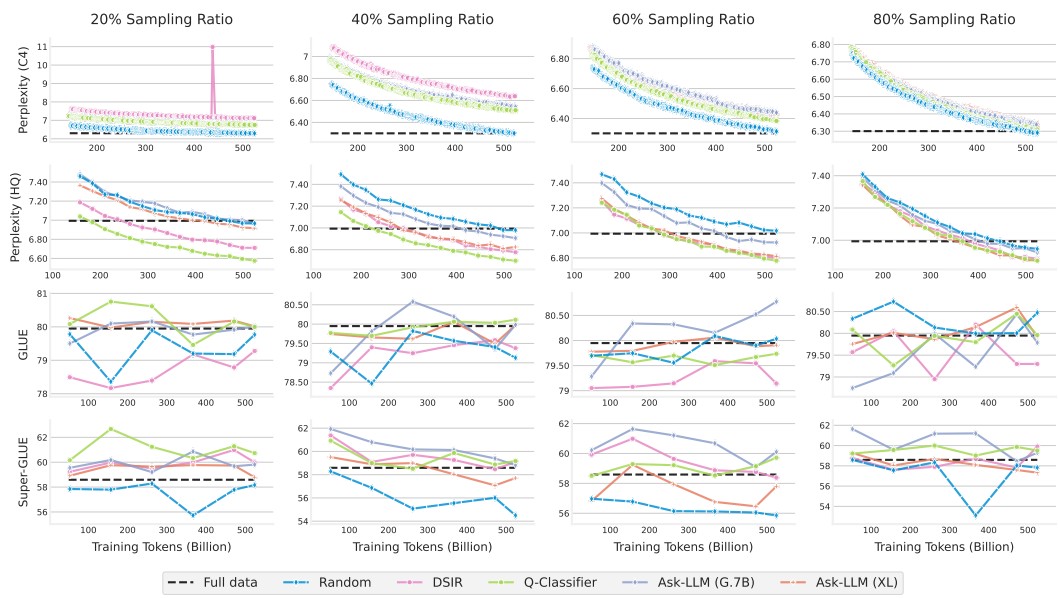

Figure 32: Data efficiency comparison of different samplers while training T5-Small for various sampling ratios. Each point in this plot is the performance of an intermediate checkpoint during the course of training on sampled data.

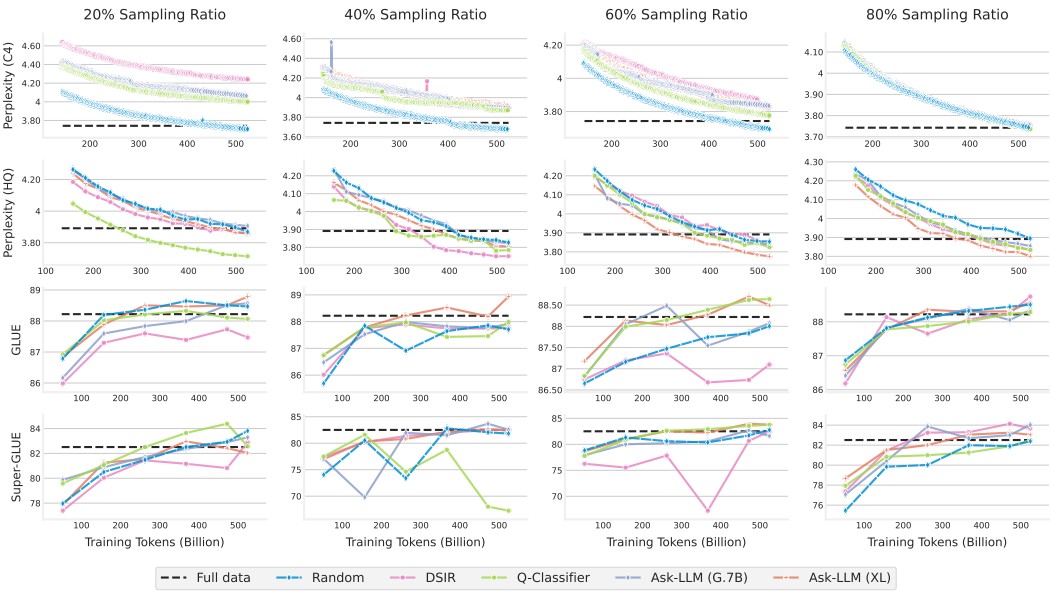

Figure 33: Data efficiency comparison of different samplers while training T5-Large for various sampling ratios. Each point in this plot is the performance of an intermediate checkpoint during the course of training on sampled data.

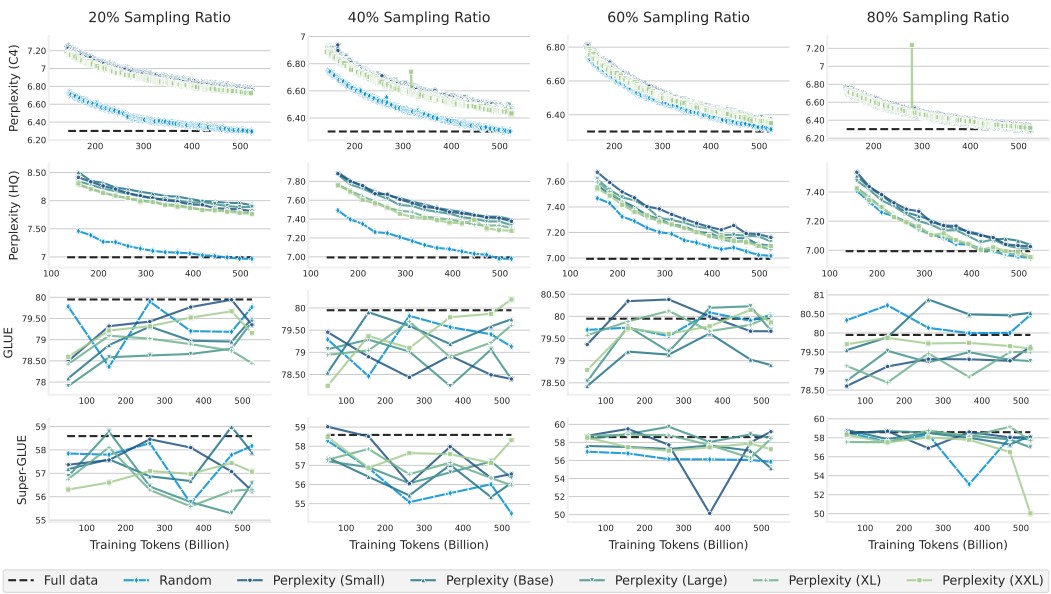

Figure 34: Data efficiency comparison of different samplers while training T5-Small for various sampling ratios. Each point in this plot is the performance of an intermediate checkpoint during the course of training on sampled data.

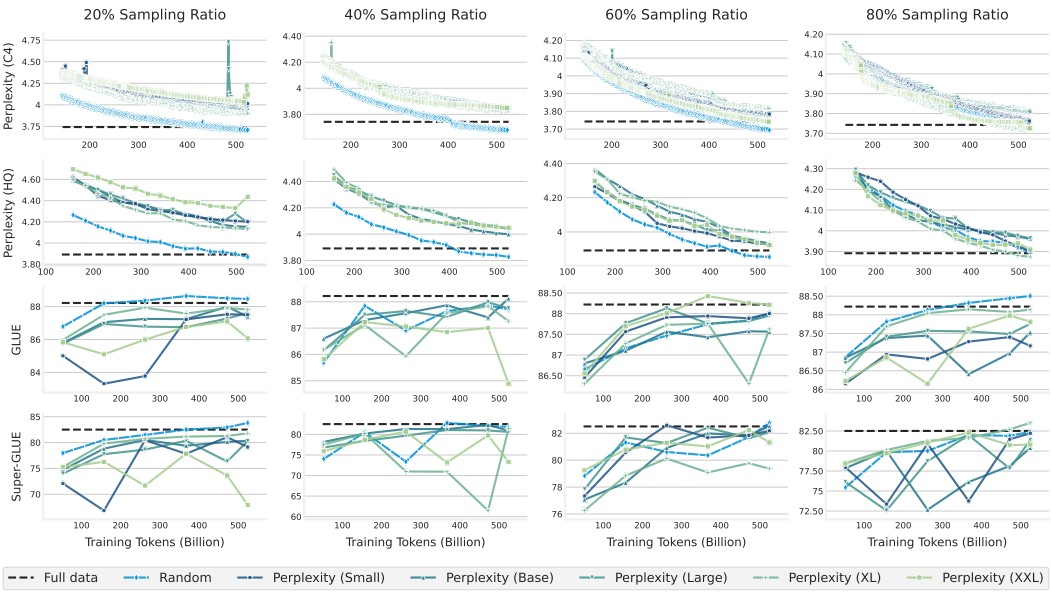

Figure 35: Data efficiency comparison of different samplers while training T5-Large for various sampling ratios. Each point in this plot is the performance of an intermediate checkpoint during the course of training on sampled data.

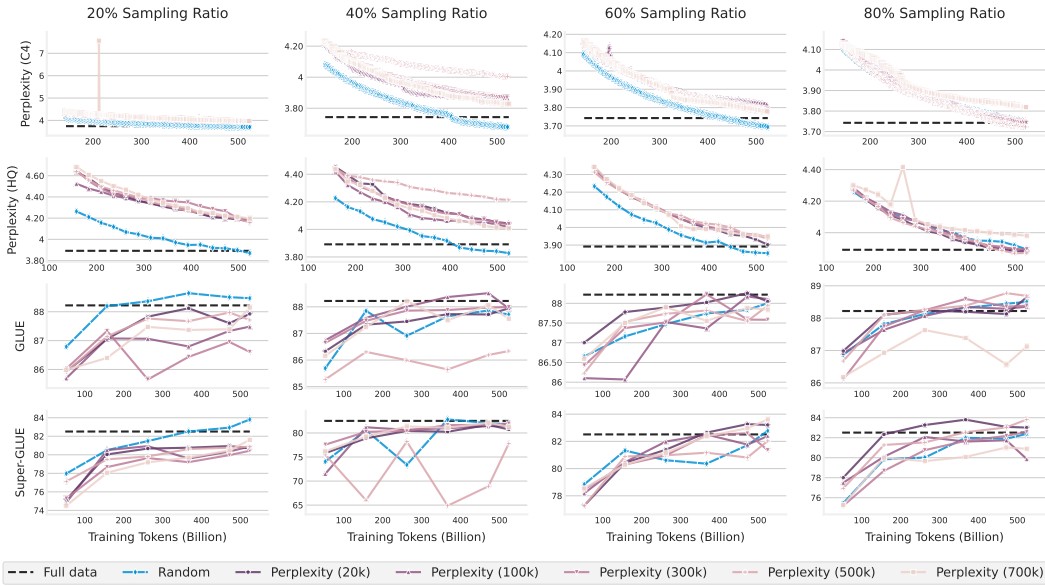

Figure 36: Data efficiency comparison of different samplers while training T5-Large for various sampling ratios. Each point in this plot is the performance of an intermediate checkpoint during the course of training on sampled data.

## H  QUALITATIVE RESULTS

In this section we look at some qualitative training samples, sorted according to various criteria of data-quality scores. Along with the textual content of each training sample, we also list the estimated data-quality percentile for ASK-LLM and perplexity filtering samplers, *i.e.*, the percentile of the given data-point's quality score amongst the entire training set. A high percentile represents that the sampler estimates this training sample to have higher quality compared to other training samples in the dataset. We manually don't include any NSFW examples to the best of our knowledge.

### H.1  HIGH-QUALITY SAMPLES IDENTIFIED BY ASK-LLM

We look at the training samples that *all* ASK-LLM scoring models, on average, think are good (*i.e.*, have a high percentile). To the best of our understanding, the overarching conclusions we make by observing these qualitative samples are:

- ASK-LLM doesn't seem to have any length bias for good examples.
- ASK-LLM can accurately tag high-quality training samples that contain a lot of proper nouns and named entities. Perplexity filtering gets these kind of samples wrong.
- Even looking at this slice of only the highest-quality data tagged by ASK-LLM, perplexity filtering scores don't seem to correlate well with ASK-LLM scores as suggested by Figure 7.

---

**Example 1: Estimated Data-Quality (Percentile – Higher is better)**

| | | ASK-LLM | | | | | Perplexity Filtering | | |
| Small | Base | Large | XL | XXL | Small | Base | Large | XL | XXL |
|---|---|---|---|---|---|---|---|---|---|
| 93.33% | 88.21% | 88.11% | 100.0% | 99.99% | 50.29% | 30.34% | 32.56% | 31.61% | 25.62% |

❧❧

What constitutes overtime for a part-time employee? Question: What is overtime for a part-time employee? Overtime for a part-time employee is time that is beyond the part-time employee's ordinary hours of work or outside the agreed number of hours of work, as specified in their employment contract.

---

> **Example 2: Estimated Data-Quality (Percentile – Higher is better)**
>
> | | Ask-LLM | | | | Perplexity Filtering | | | | |
> |---|---|---|---|---|---|---|---|---|---|
> | Small | Base | Large | XL | XXL | Small | Base | Large | XL | XXL |
> | 99.86% | 98.54% | 96.4% | 96.3% | 96.67% | 46.2% | 54.65% | 46.2% | 49.85% | 20.33% |
>
> ✿❦✿
>
> Viva La Vegan! - Can a Vegan Lifestyle Help to Get Rid of Ocean Dead Zones? Can a Vegan Lifestyle Help to Get Rid of Ocean Dead Zones? A dead zone is an area at the bottom of the ocean that is oxygen depleted and cannot maintain any marine life. The biggest cause of these dead zones is an overflow of fertilizers, sewage and industrial pollutants being pumped into rivers all over the world. Thankfully dead zones can be reversed and living a vegan lifestyle can help enormously and I'll show you how. What are Ocean Dead Zones?
> ......
> Vegans don't want to harm the planet. On the contrary they want to save it and what better way than living with nature instead of against it and helping the planet in ways we probably never even realised, like helping to reverse our oceans dead zones. Next time you think about buying something you don't need, or eating food that is highly processed or non-organic, spare a thought for the largely unknown dead zones and how overconsumption and an unnatural lifestyle is slowly killing both you and them.

> **Example 3: Estimated Data-Quality (Percentile – Higher is better)**
>
> | | Ask-LLM | | | | Perplexity Filtering | | | | |
> |---|---|---|---|---|---|---|---|---|---|
> | Small | Base | Large | XL | XXL | Small | Base | Large | XL | XXL |
> | 98.81% | 98.96% | 95.42% | 99.53% | 99.56% | 88.1% | 80.99% | 77.13% | 65.89% | 73.79% |
>
> ✿❦✿
>
> Question: Is it necessary to dredge ponds and lakes in the upper coastal region of South Carolina? Answer: It is necessary to dredge ponds and lakes in South Carolina, in the upper coastal region of South Carolina. Each lake and each pond is a different environment and as years pass, these environments accumulate a lot of sediment. They tend to fill in with storm water runoff, they tend from natural leafy materials—whether it be grass clippings, leafy materials, storm water fun off, sand, silt, sediment, muck, mire. All of these produce in the bottoms of pond beds and lake beds. So it is absolutely necessary to do an evaluation every so many years to determine whether or not you need to remove the sediment that's accumulated.

---

**Example 4: Estimated Data-Quality (Percentile – Higher is better)**

| | | ASK-LLM | | | | | Perplexity Filtering | | |
|---|---|---|---|---|---|---|---|---|---|
| Small | Base | Large | XL | XXL | Small | Base | Large | XL | XXL |
| 88.93% | 92.16% | 90.3% | 95.14% | 93.44% | 26.83% | 34.32% | 32.98% | 31.14% | 28.35% |

❧❧

However, it's a long and challenging way to mass production. New Tesla Model 3 is an electric game-changer worth $35,000 and comes in classic black color. A single masterpiece in black now belongs to Tesla's CEO and co-founder Elon Musk. Why not mass market yet? Company has a quite complicated reason. Tesla needs to make sure that it can build, deliver and service enormous numbers of these awesome electric cars without sacrificing quality.

Tesla will present 30 first cars at a launch celebration dated on July 28. 100 cars with production speed 3 cars per day dated for August. 1,500 cars will be ready for September.

...

Owners of new Teslas will also enjoy exquisite aerodynamic wheel face. An itemized list of the Tesla Model 3's features, specs, and pricing is expected to be revealed on July 28, at the car's launch party. 5.6 seconds is what it gets the Model 3 to go from zero to 60 miles per hour, as May news says. Hot, right? It accelerates even faster than the base model BMW 3 Series or the famous Mercedes-Benz C Class, which are leaders in the compact luxury space. A single charge will allow minimum 215 miles of single drive. The roof in Model 3 is made almost entirely of glass, providing an incredible sense of space and infinity. Moreover, it blocks UV rays and manages the level of heat.

---

**Example 5: Estimated Data-Quality (Percentile – Higher is better)**

| | | ASK-LLM | | | | | Perplexity Filtering | | |
|---|---|---|---|---|---|---|---|---|---|
| Small | Base | Large | XL | XXL | Small | Base | Large | XL | XXL |
| 89.28% | 98.11% | 98.93% | 98.7% | 96.32% | 26.24% | 19.14% | 26.25% | 26.05% | 24.29% |

❧❧

Landmines. Every month, 1200 people are maimed, and a further 800 killed throughout the world due to landmines. Landmine removal efforts are clearing about 100,000 mines a year, but at rate it will still be over 1000 years to get them all. The cost of clearing them is huge, with estimates in excess of $50 billion. Worse still, for every 5000 mines cleared, one person will die in the process.

...

Hopefully the work that people like Vandiver and Tan can be built upon and further progress can be made in the fight to clear the world of landmines. The video below shows a group of minesweepers working with the kits- and it is clear even watching them that the level of understanding as to how the mine operates is already improving- giving them the knowledge they need to safely diffuse any mines they encounter.

### Example 6: Estimated Data-Quality (Percentile – Higher is better)

| | | ASK-LLM | | | | | Perplexity Filtering | | |
|---|---|---|---|---|---|---|---|---|---|
| Small | Base | Large | XL | XXL | Small | Base | Large | XL | XXL |
| 87.79% | 98.52% | 90.11% | 91.65% | 88.09% | 19.72% | 17.88% | 21.13% | 16.95% | 11.92% |

By all measures a successful chemical engineering undergraduate at Oregon Agricultural College, and wanting very much to continue his education and earn his PhD in chemistry, Linus Pauling wrote to several graduate programs across the country, inquiring in particular about fellowships. Though he had proven himself to be prodigious talent as a student and, already, as a teacher, Pauling's location in Corvallis didn't carry a great deal of cache with the country's elite institutions. And given his family's shaky financial health, some measure of institutional funding was going to be required if he were to advance in the academy.

...

During his sparse free time, Pauling wrote letter after letter to his girlfriend, Ava Helen Miller, who remained in Corvallis to continue work on her Home Economics degree at OAC. Having expressed a desire to marry at least twice before Linus left for California, only to be rebuffed by their families, the two decided in their letters that they would absolutely be wed once Pauling had finished his first year of classes and just prior to his resumption of more construction work during the summer. Their plan came to fruition in Salem, Oregon on June 17, 1923, and Ava Helen moved to Pasadena that fall to accompany her new husband during his second year as a graduate student.

### Example 7: Estimated Data-Quality (Percentile – Higher is better)

| | | ASK-LLM | | | | | Perplexity Filtering | | |
|---|---|---|---|---|---|---|---|---|---|
| Small | Base | Large | XL | XXL | Small | Base | Large | XL | XXL |
| 87.08% | 89.33% | 95.26% | 99.13% | 99.94% | 98.09% | 97.52% | 98.83% | 97.39% | 97.38% |

Bonelli, N.; Giordano, S.; Procissi, G. Enif-Lang: A Specialized Language for Programming Network Functions on Commodity Hardware. J. Sens. Actuator Netw. 2018, 7, 34. Bonelli N, Giordano S, Procissi G. Enif-Lang: A Specialized Language for Programming Network Functions on Commodity Hardware. Journal of Sensor and Actuator Networks. 2018; 7(3):34. Bonelli, Nicola; Giordano, Stefano; Procissi, Gregorio. 2018. "Enif-Lang: A Specialized Language for Programming Network Functions on Commodity Hardware." J. Sens. Actuator Netw. 7, no. 3: 34.

> **Example 8: Estimated Data-Quality (Percentile – Higher is better)**
>
> | Ask-LLM | | | | | Perplexity Filtering | | | | |
> |---|---|---|---|---|---|---|---|---|---|
> | Small | Base | Large | XL | XXL | Small | Base | Large | XL | XXL |
> | 96.41% | 86.03% | 97.38% | 95.91% | 90.8% | 34.7% | 44.8% | 56.87% | 60.15% | 77.25% |
>
> ❧❧
>
> "What is your number one secret to productivity?" In recording their responses, Kruse came across some fascinating suggestions. What follows are some of my favorites. They focus on minutes, not hours. Most people default to hour and half-hour blocks on their calendar; highly successful people know that there are 1,440 minutes in every day and that there is nothing more valuable than time. Money can be lost and made again, but time spent can never be reclaimed. As legendary Olympic gymnast Shannon Miller told Kevin, "To this day, I keep a schedule that is almost minute by minute." You must master your minutes to master your life.
> ...
> Energy is everything. You can't make more minutes in the day, but you can increase your energy to increase your attention, focus, and productivity. Highly successful people don't skip meals, sleep, or breaks in the pursuit of more, more, more. Instead, they view food as fuel, sleep as recovery, and breaks as opportunities to recharge in order to get even more done. Author of #1 bestselling book, Emotional Intelligence 2.0, and president of TalentSmart, world's leading provider of emotional intelligence.

## H.2 Low-quality Samples Identified by Ask-LLM

We look at the training samples that *all* Ask-LLM scoring models, on average, think are bad (*i.e.*, have a low percentile). To the best of our understanding, the overarching conclusions we make by observing these qualitative samples are:

- Ask-LLM doesn't seem to have any length bias for bad examples.

- Ask-LLM filters hateful or toxic examples that might hurt LLM training.

- Ask-LLM rejects non-contextual samples, *e.g.*, having only questions with no answers, repeated non-sensical content, *etc.* Notably, perplexity filtering performs bad in these cases, as these low quality examples tend to have a low perplexity score.

> **Example 9: Estimated Data-Quality (Percentile – Higher is better)**
>
> | Ask-LLM | | | | | Perplexity Filtering | | | | |
> |---|---|---|---|---|---|---|---|---|---|
> | Small | Base | Large | XL | XXL | Small | Base | Large | XL | XXL |
> | 0.01% | 0.01% | 0.01% | 0.0% | 0.0% | 40.46% | 25.66% | 27.42% | 25.6% | 28.12% |
>
> ❧❧
>
> Release name : Juiced2.Hot.Import.Nights-Multi5-RELOADED. ? Format : iso Juiced 2: HIN evolves the current street racing scene, letting players experience PC Repack DiRT Rally v1.1 ? Black Box Bears Cant Drift PC torrent uploaded. ? Juiced 2 ? ? ?? ? ???? ???? ? ??? ? ?? ? ? ? ? ????! .
> ...
> HIN evolves the current street racing scene, letting players experience the culture of the real-life HIN tour, the nation?s largest lifestyle custom. Juiced 2 Hot Import Nights Torrent. Bittorrent 729.64 MB. Juiced 2 Hot Import Nights Download free torrent at Largest Bittorrent Source with Several Listed Files. Now you can upload screenshots or other images (cover scans, disc scans,...

**Example 10: Estimated Data-Quality (Percentile – Higher is better)**

| | ASK-LLM | | | | | Perplexity Filtering | | | |
|---|---|---|---|---|---|---|---|---|---|
| Small | Base | Large | XL | XXL | Small | Base | Large | XL | XXL |
| 5.41% | 3.86% | 0.49% | 0.8% | 6.24% | 62.97% | 75.91% | 86.3% | 85.26% | 88.11% |

You were a good daughter the first day or two. Now, you are only showing the worst sides of yourself. I can only be sad and disappointed in you.

**Example 11: Estimated Data-Quality (Percentile – Higher is better)**

| | ASK-LLM | | | | | Perplexity Filtering | | | |
|---|---|---|---|---|---|---|---|---|---|
| Small | Base | Large | XL | XXL | Small | Base | Large | XL | XXL |
| 1.08% | 0.41% | 6.16% | 2.46% | 1.44% | 35.97% | 24.13% | 31.46% | 51.15% | 38.19% |

Kids can help you enrich your life? Be a better person? Learn to think about someone else? Apparently whoever said these things has never had children because from everything we have seen and experienced, kids are flat out horrible. College can't come fast enough.

**Example 12: Estimated Data-Quality (Percentile – Higher is better)**

| | ASK-LLM | | | | | Perplexity Filtering | | | |
|---|---|---|---|---|---|---|---|---|---|
| Small | Base | Large | XL | XXL | Small | Base | Large | XL | XXL |
| 1.89% | 3.58% | 3.11% | 6.02% | 0.09% | 18.09% | 22.8% | 25.61% | 19.14% | 47.01% |

EventsThis is how you can go ice skating with real penguinsGrab your tickets before they sell out! Can you spot anyone you know in these fun pics? EventsHow do I get tickets for Wimbledon 2018?

**Example 13: Estimated Data-Quality (Percentile – Higher is better)**

| | ASK-LLM | | | | | Perplexity Filtering | | | |
|---|---|---|---|---|---|---|---|---|---|
| Small | Base | Large | XL | XXL | Small | Base | Large | XL | XXL |
| 2.17% | 1.11% | 3.75% | 2.0% | 5.31% | 92.49% | 89.88% | 86.79% | 97.04% | 96.78% |

That I don't make you happy? We can start all over some day? Somewhere, are you dreaming of me? Won't you come back home to me?

---

**Example 14: Estimated Data-Quality (Percentile – Higher is better)**

| | Aꜱᴋ-LLM | | | | | Perplexity Filtering | | | |
|---|---|---|---|---|---|---|---|---|---|
| Small | Base | Large | XL | XXL | Small | Base | Large | XL | XXL |
| 0.06% | 0.04% | 0.08% | 0.11% | 0.07% | 68.86% | 51.15% | 44.08% | 35.81% | 19.28% |

❀❧

? , ? , ? , ? , ? ? , ? ? . (1395). ? ? ? ? ? ? ? ? ? ? ? ? ? ? ? ? ? ? . ? ? ? ? , 26(2), 145-159. ? ?
; ? ? ; ? ? ? ? . " ? ? ? ? ? ? ? ? ? ? ? ? ? ? ? ? ? ". ? ? ? ? , 26, 2, 1395, 145-159. ? , ? , ? , ?
, ? ? , ? ? . (1395). ' ? ? ? ? ? ? ? ? ? ? ? ? ? ? ? ? ' , ? ? ? ? , 26(2), pp. 145-159. ? , ? , ? ,
? , ? ? , ? ? . ? ? ? ? ? ? ? ? ? ? ? ? ? ? ? ? ? . ? ? ? ? , 1395; 26(2): 145-159. ? ? ? ? ? ? ? ?
? ? ? ? ? BHT ? ? ? ? ? ? ? DPPH ? ? ? ? ? ? ? ? ? ? ? ? . ? ? ? ? ? ? ? ? ? ? ? ? (HPMC) ?
? ? ? ? ? ? ? ? ? ? ? ? ? ? ? ? ? ? ? ?

...

Effect of the plasticizer on permeability, mechanical resistance and thermal behaviour of
composite coating films. Powder Technology 238:14-19. Martos MV, Mohamady MA,
Fern?ndez?L?pez J, Abd ElRazik KA, Omer EA, P?rez?Alvarez JA and Sendra E, 2011. In
vitro antioxidant and antibacterial activities of essentials oils obtained from Egyptian
aromatic plants. Food Control 22: 1715?1722. Phoopuritham P, Thongngam M, Yoksan R
and Suppakul P, 2011. Antioxidant Properties of Selected Plant Extracts and Application in
Packaging as Antioxidant Cellulose?Based Films for Vegetable Oil. Packaging Technology
and Science 25: 125?136. Rojas?Gra? MA, Avena?Bustillos RJ, Olsen C, Friedman M,
Henika PR, Martin?Belloso O, Pan Zh and McHughTH, 2007. Effects...

---

**Example 15: Estimated Data-Quality (Percentile – Higher is better)**

| | Aꜱᴋ-LLM | | | | | Perplexity Filtering | | | |
|---|---|---|---|---|---|---|---|---|---|
| Small | Base | Large | XL | XXL | Small | Base | Large | XL | XXL |
| 0.01% | 0.02% | 0.02% | 0.01% | 0.0% | 59.41% | 36.81% | 23.01% | 12.95% | 17.24% |

❀❧

Showing results for tags 'A3arma_start'. I have a Error mesage "Addon 'A3_epoch_server'
requires addon 'A3_epoch_config'" why is that and how can i fix this? When i click Ok i get
this My Start.cmd losk like this: arma3server.exe [email protected];@EpochHive; -config=C:
? arma 3 ? SC ? config.cfg -ip=192.168.71.234 -port=2301 -profiles=SC -cfg=C: ? arma 3 ?
SC ? basic.cfg -name=SC This is my RPT file:

================================================================================
== C: ? arma 3 ? arma3server.exe == arma3server.exe [email protected];@EpochHive;
-config=C: ? arma 3 ? SC ?

...

2:05:23 Updating base class ->RscListBox, by a3 ? ui_f ? config.bin/RscIGUIListBox/
2:05:23 Updating base class ->RscListNBox, by a3 ? ui_f ? config.bin/RscIGUIListNBox/
2:05:23 Updating base class ->RscText, by a3 ? ui_f ? config.bin/RscBackground/ 2:05:23
Updating base class ->RscText, by a3 ? ui_f ? config.bin/RscBackgroundGUI/ 2:05:23
Updating base class ->RscPicture, by a3 ? ui_f ? config.bin/RscBackgroundGUILeft/
2:05:23 Updating base class ->RscPicture, by a3 ? ui_f ?
config.bin/RscBackgroundGUIRight/ 2:05:23 Updating base class ->RscPicture, by a3 ? ui_f
? config.bin/RscBackgroundGUIBottom/ 2:05:23 Updating base class ->RscText, by a3...

**Example 16: Estimated Data-Quality (Percentile – Higher is better)**

| | Ask-LLM | | | | Perplexity Filtering | | | | |
|---|---|---|---|---|---|---|---|---|---|
| Small | Base | Large | XL | XXL | Small | Base | Large | XL | XXL |
| 0.47% | 3.79% | 1.93% | 1.08% | 10.22% | 51.15% | 46.92% | 63.04% | 44.77% | 41.35% |

❧❦❧

10 February 2019 I have 2 houses (joint - me & my wife) in my name and 2 land (plots). Recently sold one of flat (100% cheque payment). Can I reinvest the Capital gains arriving out of sale in purchasing a flat? Note: I had reinvested earlier on (4 years ago) the similar captial gains to buy land from a house sale.

### H.3 Increasing-quality Samples Identified by Ask-LLM

We look at the training samples that Ask-LLM scoring models *disagree on* as we go from Flan-T5-Small → Flan-T5-XXL. Specifically, we look at training samples that Flan-T5-Small thinks are of low quality, whereas Flan-T5-XXL thinks otherwise. To the best of our understanding, our overarching conclusions by observing these qualitative samples are:

- Larger scoring models in Ask-LLM are able to identify training samples containing *tail-end* of knowledge, *e.g.*, rare world-events, rare named entities, *etc.*

- The increasing quality trend going from Flan-T5-Small → Flan-T5-XXL isn't correlated with the quality scoring model size in perplexity filtering.

**Example 17: Estimated Data-Quality (Percentile – Higher is better)**

| | Ask-LLM | | | | Perplexity Filtering | | | | |
|---|---|---|---|---|---|---|---|---|---|
| Small | Base | Large | XL | XXL | Small | Base | Large | XL | XXL |
| 7.67% | 30.45% | 57.41% | 78.17% | 97.41% | 15.56% | 31.02% | 24.14% | 50.59% | 49.64% |

❧❦❧

The historic city of Manchester now features one of the most interesting public art installations that art lovers have ever witnessed. Design studio, Acrylicize installed five giant lamps in Piccadilly Place that represent the many historic periods that the city has gone through, including; Art Deco, Art Nouveau, Victorian, mid-century, and contemporary. The installation is without any doubt, a great piece of art but unlike other artworks, these are absolutely functional as well. Each lamp provides the many visitors with seating, shelter, light and even heat in the winters. The admirers can also witness the historic stories of Manchester via graphic illustrations on the lamps.

**Example 18: Estimated Data-Quality (Percentile – Higher is better)**

| | Ask-LLM | | | | Perplexity Filtering | | | | |
|---|---|---|---|---|---|---|---|---|---|
| Small | Base | Large | XL | XXL | Small | Base | Large | XL | XXL |
| 10.48% | 31.26% | 54.17% | 84.17% | 97.93% | 30.52% | 39.49% | 35.79% | 30.89% | 25.39% |

❧❦❧

The Cokin Yellow and Pink Center Spot filter has a clear center and diffused yellow and pink edges. Theses diffused edges will be produce blur while leaving the center sharp. The filter effect is directly influenced by the f-stop and the focal length. A lens shot at f/1.4 will see a greater blurring effect than f/8.0 and a 85mm lens will see more blur than a 28mm. Additionally, a longer focal length lens will visually increase the size of the center spot area because it sees less of the filter area.

Example 19: Estimated Data-Quality (Percentile – Higher is better)

| ASK-LLM | | | | | Perplexity Filtering | | | | |
| Small | Base | Large | XL | XXL | Small | Base | Large | XL | XXL |
| --- | --- | --- | --- | --- | --- | --- | --- | --- | --- |
| 7.05% | 20.29% | 38.23% | 50.38% | 63.94% | 22.41% | 14.8% | 12.69% | 20.68% | 8.62% |

Provide hoist coverage and 200 degree rotation for individual use in bays, along walls, or columns of plants, or as a supplement to an overhead crane or monorail system. This jib has the advantage of providing maximum lift for the hoist, since it can be installed very close to the underside of the lowest ceiling obstruction. It is composed of a vertical mast mounted to 2 brackets on a wall or vertical building beam with a boom that cantilevers out, perpendicular from the wall at the top.

Example 20: Estimated Data-Quality (Percentile – Higher is better)

| ASK-LLM | | | | | Perplexity Filtering | | | | |
| Small | Base | Large | XL | XXL | Small | Base | Large | XL | XXL |
| --- | --- | --- | --- | --- | --- | --- | --- | --- | --- |
| 20.76% | 45.81% | 60.22% | 73.95% | 84.14% | 2.98% | 2.94% | 3.49% | 2.51% | 2.09% |

The mighty Adyar River that flows through Chennai has a tale to tell. Arun Krishnamurthy, founder, Environmentalist Foundation of India has documented the origin of the river, the journey and the culmination all captured in images aimed at sensitizing citizens of Chennai to a treasure that they are being denied. Titled Urban Waters, the photo exhibition on Adyar river will bring out Adyar's rich history, fine ecology, urban exploitation and her innate beauty through framed images. The exhibition is organised at Max Mueller Bhavan in Chennai. Goethe Institut, Max Mueller Bhavan is at 4, 5th Street, Rutland Gate, Chennai.

Example 21: Estimated Data-Quality (Percentile – Higher is better)

| ASK-LLM | | | | | Perplexity Filtering | | | | |
| Small | Base | Large | XL | XXL | Small | Base | Large | XL | XXL |
| --- | --- | --- | --- | --- | --- | --- | --- | --- | --- |
| 4.27% | 22.22% | 47.57% | 82.58% | 92.4% | 6.34% | 4.77% | 3.89% | 8.75% | 7.55% |

The Pendaries Village Skyline Subdivision is located near both the Santa Fe National Forest and the Pecos Wilderness in North Central New Mexico. It has the charm of small town New Mexico, perhaps even more so than its better known nearby sister cities. It offers a unique opportunity for people wishing to enjoy the quiet beauty of Northern New Mexico.

**Example 22: Estimated Data-Quality (Percentile – Higher is better)**

| | Ask-LLM | | | | | Perplexity Filtering | | | |
|---|---|---|---|---|---|---|---|---|---|
| Small | Base | Large | XL | XXL | Small | Base | Large | XL | XXL |
| 22.09% | 66.57% | 76.56% | 85.51% | 96.98% | 20.8% | 24.82% | 17.42% | 18.65% | 15.55% |

Anderson .Paak's new album, Oxnard, is a nod to the Southern California city where Anderson grew up. It is the Grammy-nominated artist's third studio album and the first to be released on Dr. Dre's label Aftermath Entertainment. Oxnard includes his latest single, Tints featuring Kendrick Lamar along with album features from J Cole, Pusha T and many more. This is the album he dreamed of making in high school, when he was listening to Jay-Z's The Blueprint, The Game's The Documentary, and Kanye West's The College Dropout. The classic fourth album from the rap-god Eminem.

**Example 23: Estimated Data-Quality (Percentile – Higher is better)**

| | Ask-LLM | | | | | Perplexity Filtering | | | |
|---|---|---|---|---|---|---|---|---|---|
| Small | Base | Large | XL | XXL | Small | Base | Large | XL | XXL |
| 0.98% | 24.84% | 53.36% | 88.98% | 98.18% | 2.3% | 1.48% | 2.03% | 2.1% | 3.07% |

The Disknet is a networking solution which uses the external floppy drive port of the Amiga. It uses the same coax cabling as 10Base2 Ethernet (RG-58U/50Ohm) but is NOT compatible and is capable of transferring at around 45k/sec. The Disknet may be the same device as the AmigaLink, but this has not been confirmed.

## H.4 Decreasing-quality Samples Identified by Ask-LLM

We look at the training samples that Ask-LLM scoring models *disagree on* as we go from Flan-T5-Small → Flan-T5-XXL. Specifically, we look at training samples that Flan-T5-XXL thinks are of low quality, whereas Flan-T5-Small thinks otherwise. To the best of our understanding, our overarching conclusions by observing these qualitative samples are:

- Smaller quality-scoring models sometimes mislabel non-informative training samples, that contain, *e.g.*, non-informative content, or repeated content.
- The decreasing quality trend going from Flan-T5-Small → Flan-T5-XXL isn't correlated with the quality scoring model size in perplexity filtering.

**Example 24: Estimated Data-Quality (Percentile – Higher is better)**

| | Ask-LLM | | | | | Perplexity Filtering | | | |
|---|---|---|---|---|---|---|---|---|---|
| Small | Base | Large | XL | XXL | Small | Base | Large | XL | XXL |
| 64.05% | 46.39% | 35.92% | 25.29% | 9.63% | 4.3% | 10.21% | 3.47% | 3.34% | 3.35% |

one filled with goodwill and cheer. who have supported me thru the year. I wouldn't be changing careers. instead of on strange people's rears. Wishes You a Healthy, Happy Holidays! Ah, how the mighty have fallen! And a Merry fave to you ... and a happy new rear. From one Xmas humor story to another, enjoyed this! Thanks Jack & Susan! Doug, I checked him out–wonderful stuff! Will pass along the good word. Fun and funny–as always! Thanks for the cheer! I can only fave this once, but I've looked at it repeatedly over what has been a bizarre week– and each time you've given me a laugh. That's a gift Bob and I'm grateful! Best of holidays to you and a great New Year!

> **Example 25: Estimated Data-Quality (Percentile – Higher is better)**
>
> | | ASK-LLM | | | | | Perplexity Filtering | | | |
> |---|---|---|---|---|---|---|---|---|---|
> | Small | Base | Large | XL | XXL | Small | Base | Large | XL | XXL |
> | 91.25% | 71.8% | 53.1% | 24.11% | 4.53% | 32.4% | 36.56% | 46.53% | 48.19% | 54.84% |
>
> I hear people saying that vinyl records have a better sound quality than CDs or even DVDs. A mini LP is a CD version of something that was originally released as a 12" (12 inch) vinyl LP. In many cases the packaging is superior to, or at least. Vitalogy; Studio album by Pearl Jam; Released: Vinyl: November 22, 1994 CD: December 6, 1994: Recorded: November 1993 – October 1994: Studio: Bad Animals Studio. Browse best sellers, new releases, AutoRip CDs and vinyl records, deals, vinyl Audio CD. 7.99. From A Room: Volume 1. Chris Stapleton. Audio. The one and only CD, DVD, VIDEO, DJ, VINYL, ERO store. Search our full catalog. Recordstore.co.uk. The UK's leading online record store. Buy new and exclusive signed bundles, CDs, LPs, Merchandise and box sets. Recordstore Day, every. Vinyl Records to CD Conversion - Cheapest on the net! High-quality, standards-compliant CD-Audio of your favorite vinyl records, saved for posterity. Custom CD, DVD Vinyl Packaging You're just a click away from a gorgeous, retail-ready CD or DVD in professional disc packaging. We also offer a full-range of Vinyl.
> ...
> Buy with confidence as the. Mar 4, 2017 Despite the decline in mainstream CD usage, some consumers still have CD recording needs for radio, vinyl and other formats. Here are our. 12 results . You can finally burn your cassettes and vinyl records to CD with Crosley's Memory Master II CD Recorder. Just play your cassette or record One Nation is back after the Sold Out New Years Eve event with yet another From its esoteric origins releasing field recordings of steam engines on vinyl to our latest critically acclaimed Ultradisc UHR™ SACDs, Mobile Fidelity Sound. How much are worth and valued your rare and collectable vinyl and cd by searching on Music Price Guide archive. Heel veel CD, LP, Vinyl SACD op voorraad, snelle levertijden en altijd superscherp geprijsd en lage verzendkosten, voor 17:00 besteld morgen Some of the greatest music ever made isn t available digitally, on mp3, or on CD; but rather is only available on vinyl. Moreover, if you already have purchased.

**Example 26: Estimated Data-Quality (Percentile – Higher is better)**

| | Ask-LLM | | | | Perplexity Filtering | | | | |
|---|---|---|---|---|---|---|---|---|---|
| Small | Base | Large | XL | XXL | Small | Base | Large | XL | XXL |
| 96.67% | 76.07% | 47.33% | 30.0% | 7.97% | 32.02% | 21.27% | 24.31% | 25.77% | 23.7% |

❧❧

A brilliant performance by Year 6 based on The Lion King. Brilliant singing and acting from everyone, congratulations Year 6! A big thank you to all the staff that helped with everything from costumes, set design, make up and directing. A wonderful commemoration of the seven years that Year 6 students have spent at The Good Shepherd. Thank you to all of the parents and staff for attending this celebration and we wish all of the children continued success in their new schools and hope they continue to do themselves proud. Well done to Foundation for showing us what it is to be good friends! This week we have been looking at all the countries in the world that speak Spanish as their native language, there are 21! So throughout school we spent a day learning lots of wonderful things about our chosen country. We looked at maps, flags, famous people, food and so much more! Below is a little glimpse into our fabulous week.

...

Click on the links to take a look at some of the brilliant things we got up to! Faith in Families is a charity based here in Nottingham who believe, as we do, that all children have the right to grow up as part of a loving and nurturing family and they provide services for children and families. We learnt lots about adoption and what it can mean for children and their family. We learnt about Fairtrade and all the fantastic work they do around the world. We also discovered lots of products that we did not know were Fairtrade. There was also a sell out Fairtrade food sale, well done everyone! Year 2 have been able to show off our brilliant new high visibility jackets! Now we will be able to stay safe and visible on any out of school trips. We are very lucky to have these donated by Walton & Allen. Thank you! Click on the high visibility jacket to take a look at our super jackets! Year 4 have wowed us with their acting skills in a brilliant performance of Ali Baba - well done Year 4! Year...

---

**Example 27: Estimated Data-Quality (Percentile – Higher is better)**

| ASK-LLM | | | | | Perplexity Filtering | | | | |
|---|---|---|---|---|---|---|---|---|---|
| Small | Base | Large | XL | XXL | Small | Base | Large | XL | XXL |
| 90.79% | 75.97% | 58.89% | 18.06% | 3.0% | 13.65% | 16.88% | 17.85% | 14.36% | 13.67% |

❧❧

Search result for " For Sale " We supply Germany made embalming powder in small quantities from 1 kg at affordable prices. We have white and pink 100% hot and 98% pink in stock. Call us on +27786893835 for details. EMBALMING.. EMBALMING POWDER CALL +27786893835 Hager Werken Embalming Compound Pink Powder call +27786893835 in General items from Germany Embalming compound in powder form both PINK and WHITE Radio active.. Sierra Residences Type B, Sg Ara near PISA, Factory,Air-port Sierra Residences (ID: 5695) ================== Monthly Rent: RM 1,000 BU: 1182 sq.ft. Newly Renovated/NOT Furnished - 3.. Very Strategic and Highly Potential LAND 9.7 Acres Converted Residential Land For Sale in Taman Melawati !!!!! Taman Melawati development land , Titile : Freehold, non bumi land. Status:.. I am a Certified Private Loan Lender, Do you need a Fast and Guarantee loan to pay your bills or start up a Business? I offer both local and international loan services to meet your financial needs..

...

Introducing our mining company to you for a very fruitful business transaction. we are a miners who have come together to upgrade our production through the introduction of modern technology and.. Commercial land for sale. Location near to Premium Outlet. Size = 32 acres Good land shape and very suitable for development. Selling price RM 60 per sf. Interested party kindly contact.. Keterangan : * Tanah yang rata dan sangat startegik untuk buat rumah kediaman/rumah rehat (homestay), atau untuk rumah penginapan sendirian/Percutian (vacation home) * Tanah lot tepi berdekatan.. Limited gated Semi D at Sri petaling,fully furnish with lift and move in condition.newly buit,modern,spacius and practical.Prime location for own stay,good gated security and easy access to few main.. Land for sale in MELAKA ! Price : RM 65 per sq fit (or roughly U$D 17 per sq fit ) Size : 53000 sf Property type ï¼šfreehold housing land Location : Jalan Laksamana Cheng Ho,Â ..

Example 28: Estimated Data-Quality (Percentile – Higher is better)

| | Ask-LLM | | | | | Perplexity Filtering | | | |
|---|---|---|---|---|---|---|---|---|---|
| Small | Base | Large | XL | XXL | Small | Base | Large | XL | XXL |
| 94.72% | 87.31% | 78.07% | 13.77% | 6.51% | 5.75% | 9.63% | 13.12% | 17.51% | 17.12% |

❧❦

FIFA 20 CONFIRMED TRANSFERS SUMMER 2019 & RUMOURS | w/ ALEX SANDRO BALE & NEYMAR JR. TO BARCELONA!! Top 10 Worst Transfers In Football History! 70 CONFIRMED TRANSFERS JANUARY 2019 ———————————— Thank You For Watching ———————————————— * Like + Subscribe * ==================. FIFA 20 | CONFIRMED TRANSFERS SUMMER 2019 & RUMOURS | w ZIDANE COUTINHO & RONALDO BACK TO R.MADRID! REBUILDING REAL MADRID | DREAM TEAM LINEUP 2019-2020 | POTENTIAL TRANSFERS | w/ NEYMAR & RONALDO! FIFA 20 | CONFIRMED TRANSFERS SUMMER 2019 & RUMOURS | w BALE FEKIR UMTITI & NEYMAR £300M TO MADRID! SUBSCRIBE http://bit.ly/SoccerAMSub Dean from 442oons is back with his list of the top 5 deals that were done on transfer deadline day. Do you agree with .. FIFA 20 | CONFIRMED TRANSFERS SUMMER 2019 & RUMOURS | w STERLING JAMES AUBAMEYANG & GRIEZMANN! SUBSCRIBE to FOOTBALL DAILY: http://bit.ly/fdsubscribe Last week we broke down our best signings of the summer so far. Now lets expose the worst! Top 150 confirmed transfers / signings of the summer transfer window 2018 ft. Ronaldo, Mbappe, Mahrez, Vidal, Courtois... THANK FOR WATCHING! FIFA 20 | CONFIRMED TRANSFERS SUMMER 2019 & RUMOURS | w/ POGBA SANCHO THIAGO & MESSI TO INTER!!

Example 29: Estimated Data-Quality (Percentile – Higher is better)

| | Ask-LLM | | | | | Perplexity Filtering | | | |
|---|---|---|---|---|---|---|---|---|---|
| Small | Base | Large | XL | XXL | Small | Base | Large | XL | XXL |
| 86.25% | 69.2% | 61.9% | 46.57% | 19.99% | 76.61% | 71.91% | 94.86% | 92.93% | 94.99% |

❧❦

Phone 1300 616 202 if you're looking for a trustworthy, experienced and licensed Plumber Leopold. We know that getting plumbing repairs in Leopold can be a pain and you've got better things to do than look for a plumber. Clearwater Plumbing and Maintenance will save you from any unnecessary hassle and expense for a Plumber Leopold. We make sure that wherever you need a Plumber Leopold, Clearwater Plumbing and Maintenance will assist you with your plumbing worries. Plumbing problems with your taps, toilets, gas, hot water and drains are painful enough. You don't need the extra stress of finding a Plumber Leopold that you can trust. And what about all of those plumbers in Leopold who don't clean up after themselves, leaving mud and materials all over your home? Our professional team are different!

...

Do you have hot water system repairs Leopold. We have highly experienced plumbers who know how to fix hot water systems Leopold. There can be many possible reasons why your hot water system Leopold is broken. Our Leopold plumbers are reliable, fast and know hot to diagnose problems. Our hot water system repairs Leopold plumbers are trained and qualified. To book an appointment, please call 1300 616 202. We will do our best to get a plumber to you in Leopold as soon as possible. If you notice that there is water leaking from the bottom of your hot water system in Leopold, chances are the system is completely broken. In this scenario, you will need to replace your hot water system in Leopold. Our team of plumbers can help you to choose what hot water system you will need.

**Example 30: Estimated Data-Quality (Percentile – Higher is better)**

| | Ask-LLM | | | | | Perplexity Filtering | | | |
|---|---|---|---|---|---|---|---|---|---|
| Small | Base | Large | XL | XXL | Small | Base | Large | XL | XXL |
| 82.64% | 75.2% | 63.2% | 29.51% | 8.94% | 78.34% | 82.07% | 91.01% | 87.78% | 88.02% |

❧❧

You can now configure the minimum TLS protocol level for client connections and connections to other servers. Refer to the following page for more information: Advanced TLS. You can now set an Integrated Capture Point (ICP) to stopped mode by changing the state of the corresponding configuration object to disabled; changing the state to enabled restarts the inbound cycle of the ICP. You can now set the minimum TLS protocol level for the Web Service Capture Point by configuring the option <sec-protocol> in the section <settings> of the Capture Point object.

...

Support for the following databases. See the Supported Operating Environment: eServices page for more detailed information and a list of all supported databases. No special procedure is required to upgrade to release 8.5.201.05. Retrieved from "https://docs.genesys.com/Documentation:RN:mm-ixn-svr85rn:mm-ixn-svr8520105:8.5.x (2019-04-21 22:59:48)" This page was last modified on November 8, 2018, at 08:48.

**Example 31: Estimated Data-Quality (Percentile – Higher is better)**

| | Ask-LLM | | | | | Perplexity Filtering | | | |
|---|---|---|---|---|---|---|---|---|---|
| Small | Base | Large | XL | XXL | Small | Base | Large | XL | XXL |
| 62.21% | 54.71% | 35.73% | 22.64% | 6.76% | 64.82% | 85.95% | 94.65% | 93.35% | 85.29% |

❧❧

are willing to provide you with perfect services and striding for Display Stand For Boutique , Display Stand for Boutique , Display Stand for Phone , Our product quality is one of the major concerns and has been produced to meet the customer's standards. "Customer services and relationship" is another important area which we understand good communication and relationships with our customers is the most significant power to run it as a long term business. "We have quite a few great team customers very good at internet marketing, QC, and dealing with kinds of troublesome trouble while in the output approach for Display Stand For Boutique , Display Stand for Boutique , Display Stand for Phone , We set a strict quality control system. We've got return and exchange policy and you can exchange within 7 days after receive the wigs if it is in new station and we service repairing free for our solutions. You should feel free to contact us for further information and we are going to give you competitive price list then.

