# OpenReview forum: "How to train data-efficient LLMs"
_ICLR.cc/2026/Conference — ICLR 2026 Poster_

### Official Review · Reviewer_THC1 · 2025-10-30

**Soundness:** 4
**Presentation:** 3
**Contribution:** 4
**Rating:** 8
**Confidence:** 2

**Summary:**

This paper explores how to balance data coverage and model quality in large language model training. The goal is to find sampling strategies that minimize the amount of data needed while still maximizing model performance, essentially pushing the Pareto frontier of efficiency.

**Strengths:**

1.The study is timely and relevant, offering solid empirical evidence that directly contributes to the ongoing conversation on efficient LLM training.

2.The experiments are extensive — covering a wide range of model sizes (from 60M to 800M parameters) and 22 different sampling strategies — giving the conclusions strong credibility.

**Weaknesses:**

1.All experiments are conducted on T5-style encoder–decoder architectures, with no validation on the more widely used decoder-only models (like GPT). This limits the generalizability of the results.

2,Although ASK-LLM demonstrates efficiency, it still requires scoring every single sample in the dataset, which makes it computationally expensive. The practical scalability and real-world deployment cost are not fully addressed.

**Questions:**

1.The ASK-LLM approach is innovative — using the model itself to assess data quality rather than relying on external heuristics — but it may risk creating a “self-reinforcing bubble,” where the model’s own biases limit data diversity.

2.The ASK-LLM method is computationally costly. It might be worth exploring hybrid strategies, such as integrating active learning to selectively annotate boundary samples and iterate.

3.While the empirical evidence is rich, the paper would benefit from a clearer theoretical explanation of why ASK-LLM captures data quality better than traditional perplexity-based filtering.

---

> ### Author Response · Authors · 2025-12-02
>
> Thank you for taking the time to read our paper and for your positive review. We address the weaknesses and questions (in order) below.
>
> 1. **Decoder-only models.** We can report that a variation of AskLLM was used successfully in the training of the Gemma 3 series of models (briefly mentioned in our paper). The main differences lie in the prompt and in how the score is used (data re-weighting rather than filtering). We unfortunately cannot provide detailed ablations with decoder-only architectures that compare different methods of quality ranking in a controlled setting.
>
> 2. **Cost.** To answer your question, we mined our experiment logs for the accelerator-hour costs of scoring and training on C4. Note that our jobs ran at a low priority on a shared compute system and were preempted frequently. We use the accelerator duty cycle to estimate the time that it was in use, but this is an approximation because training and inference have different bottlenecks and can even occur on different hardware (for this data, we restrict our comparison to jobs on the same hardware type). We report the average cost over 30 training runs.
>
> | Scoring Model | Accelerator-Hr |
> | - | - |
> | T5-XXL | 49 |
> | T5-XL | 10 |
> | T5-Large | 1.7 |
> | T5-Base | 0.76 |
> | T5-Small | 0.24 |
>
> | Training Model | Accelerator-Hr |
> | - | - |
> | T5-Large | 24 |
> | T5-Small | 9.3 |
>
> Figures 1 and 4 suggest that, conservatively, a 44% reduction is possible when training T5-L on data selected by T5-XL. This corresponds to 56% $\times 24 + 10 \approx 23.44$ accelerator-hours with AskLLM compared to 24 accelerator-hours without AskLLM, with the benefit that the scores can be reused. Of course, these numbers are approximate, but the takeaway is that the costs are reasonable - especially as cheaper scoring models still have some benefit. We also note that AskLLM is cheaper than some techniques that are already used at scale (e.g., the logit-level distillation used in Gemma 3).
>
> 3. **Self-reinforcing bubble.** This is a very good question. AskLLM displays a smaller distribution bias than perplexity filters because the instruction-following behavior of the LLM judge overrides the familiarity bias that causes language models to output higher-confidence scores for data similar to their training distribution. However, LLMs do still have biases even when prompted in this way. We examine this issue rigorously in Appendix G (see Figure 11), finding that different LLM judges exhibit strong preferences for some topics over others. One way to address this issue is to enforce diversity separately, e.g., by applying LLM filtering at a per-topic level or by using a clustering process similar to the one described in Nvidia’s recent “CLIMB” paper.
>
> 4. **Hybrid strategies.** We agree that reducing the AskLLM scoring cost is an important direction, and active learning is a good method to reduce the number of annotations. Another good direction is to reduce the cost of each annotation. Our ablations show reasonable performance from T5-Base versus T5-XL, even though the former is more than 10x cheaper to run, suggesting that we may be able to achieve cost-effective scoring by improving the performance of these smaller models on the quality labeling task.
>
> 5. **Theory.** While not our main focus, theory can provide some simple intuition about why AskLLM can outperform perplexity filters. The paper *“Data pruning and neural scaling laws: fundamental limitations of score-based algorithms”* (TMLR 2023) contains a very good analysis of a general class of score-based algorithms that includes both AskLLM and perplexity filtering. The high-level intuition of their “no free lunch” theorem is that one can always construct an adversarial eval out of exactly the points that are discarded by the sampling algorithm.
>
> A perplexity filter simply estimates the likelihood that an example came from the same distribution that was used to train the perplexity-scoring model, i.e., $\mathrm{Pr}[x ~ D]$, and any out-of-distribution capability is likely to score poorly. AskLLM instead uses a conditional probability that depends on the scoring rubric as well as the dataset used to train the LLM-judge, i.e. $\mathrm{Pr}[\mathrm{Yes} | x, R, D]$. This conditioning process modulates the probability with the rubric and assigns scores in a less distribution-dependent way. Examples 4-6 in Appendix H.1 show how this can increase the score relative to perplexity filtering, and Examples 10,13 in Appendix H.2 show how it can decrease.
>
> Once again, thank you for your positive review. We understand that further discussion is not possible due to the recent OpenReview bug / anonymization leak, but we do hope that you feel we have addressed your concerns. We are happy to add a summarized form of this discussion to the main text, and we also noticed some formatting problems in the paper while responding to this review, which we will fix in the revision.

---

### Official Review · Reviewer_72fC · 2025-10-30

**Soundness:** 2
**Presentation:** 3
**Contribution:** 2
**Rating:** 6
**Confidence:** 4

**Summary:**

This paper explores how to efficiently train Large Language Models (LLMs), with its core focus on investigating the impact of data selection on pre-training efficiency and model quality. It proposes two new data sampling methods:

#### ASK-LLM (Quality-Based)
This approach that leverages the zero-shot reasoning capability of an existing, instruction-tuned LLM (a "proxy model"). Through specific prompts, it "asks" this proxy model to directly evaluate whether each training sample is "information-rich and high-quality."

#### DENSITY (Coverage-Based)
This method aims to maximize data diversity. It models the data distribution and prioritizes selecting samples from regions that are "under-represented" in the distribution.


The paper conducts large-scale empirical research: it tests up to 22 different data curation techniques on the T5 model, involving hundreds of pre-training runs and thousands of fine-tuning evaluations.

It draws key conclusions regarding "quality" vs. "coverage":
- Coverage-based sampling (e.g., DENSITY) can usually "recover" or "match" the performance achieved by training with the full dataset.
- Quality-based sampling (e.g., ASK-LLM) is capable of "surpassing" the performance of training with the full dataset.

Experiments show that even when using only 10% of the original data, ASK-LLM trains models that outperform those trained on 100% of the data—and with a 70% faster convergence rate.

**Strengths:**

The greatest strength of this paper lies in its proposal of the innovative ASK-LLM method.

Instead of adhering to traditional perplexity or classifier scoring, it pioneers the use of the zero-shot reasoning capability of an existing LLM. Through specific prompts, it "asks" the model to directly determine whether a data point contains "informational signals."
ASK-LLM explicitly resolves several key flaws of traditional perplexity filtering:
- **Avoiding meaningless content**: Perplexity filtering tends to select high-frequency, repetitive "nonsense" (e.g., "Example 14" and "15"), while ASK-LLM can identify such content as low-quality.
- **Preserving context**: Perplexity filtering incorrectly selects context-deficient fragments like "questions without answers," whereas ASK-LLM can properly recognize these as non-"informational" data.
- **Retaining "long-tail knowledge"**: Perplexity filtering mistakenly discards valuable yet "niche" knowledge containing rare words (e.g., "Example 17"), but ASK-LLM can leverage its reasoning ability to identify and retain such "long-tail knowledge."

The paper features an impressive volume of experiments, making its conclusions highly credible.
The paper compares up to 22 different data curation techniques.  To complete this comparison, the researchers conducted 220 independent pre-training runs and 1,100 downstream task fine-tuning evaluations. In the field of LLM pre-training, such a large-scale comparison is extremely rare and valuable.
Experiments were conducted on two model sizes (T5-Small and T5-Large) and five data sampling rates (10% to 80%), ensuring the robustness of the conclusions.

The paper clearly answers the core question raised in the introduction: "Which matters more, Quality or Coverage?"
- The upper limit of coverage-based sampling (e.g., DENSITY) is to **"recover" or "match"** the performance of full-data training.
- Quality-based sampling (e.g., ASK-LLM) can **"surpass"** the performance of full-data training.

The paper does not stop at superficial results but provides in-depth analyses:
- **Correlation Analysis (Figure 7)**: By calculating the correlation of scores from different samplers, the paper proves that ASK-LLM, Perplexity, and DENSITY indeed select distinctly different data—confirming that ASK-LLM represents a brand-new evaluation dimension.
- **Qualitative Analysis (Appendix H)**: It provides numerous real cases of data selected or discarded by ASK-LLM, intuitively demonstrating why it outperforms perplexity filtering.
- **Scoring Model Scaling (Figure 6)**: The paper finds that using a more powerful LLM (from Small to XXL) as the "judge" for ASK-LLM leads to better data filtering results. This indicates that the ASK-LLM method itself has excellent scalability and can become more effective in the future as "judge" models iterate.

**Weaknesses:**

1. Concerns Regarding Model/Data Scale and Method Generalizability (Scaling & Generalizability)

Model Scale
T5-Large (800M parameters) is indeed hardly qualified to be called a "Large Language Model" (LLM) today—even in the 2023-2024 period. Core challenges and phenomena in pre-training (such as emergent abilities) are most pronounced primarily in models with tens of billions or even hundreds of billions of parameters. Therefore, it is questionable whether the "optimal" data strategy derived from an 800M-parameter model can be directly generalized to models with 70B or 1T parameters.

Data Quality and Generalizability
C4 is a relatively "noisy" corpus that has not undergone fine-grained filtering. For such datasets, the room for improvement through "data quality filtering" is inherently large (a scenario akin to "picking the best from a bad lot").

This gives rise to a risk: the remarkable success of ASK-LLM—reducing data volume by 90%, cutting training time by 70%, while achieving better performance—may overestimate the method’s marginal benefits when applied to cleaner, higher-quality datasets. Examples of such datasets include RefinedWeb, Dolma, or the "textbook-level" data used for Phi-2. If a dataset is already of high quality, how much additional improvement can ASK-LLM bring? This is a question the paper cannot answer.

2. Another issue is that the related works cited in the paper are relatively outdated, with a lack of recent relevant studies. This makes it difficult to assess whether the paper still holds innovativeness.

**Questions:**

1. What is the true cost-benefit trade-off?
In Section 6 and Appendix C.1, the paper brilliantly argues that the high scoring cost of ASK-LLM is a "one-time amortizable" investment.

My question is: Given today’s dataset scales of 10T+ tokens, using a powerful proxy model (7B+ parameters) to perform a full inference on every training sample makes this "one-time" cost extraordinarily high.

The paper claims that training speed is accelerated by 70%, but if the scoring process itself requires computational resources equivalent to (or even exceeding) the total training time, what would the net benefit of this method be in practice?

I would very much like to see a more direct cost analysis: (scoring cost + 70% of training cost) vs. (100% of training cost). The paper fails to provide this critical comparison.


2. What exactly is the "Gemma 3 variant"?
The paper’s strongest (and most clever) response to concerns about "scalability" comes in Section 6, where it states: "...a variant of our ASK-LLM technique was used in the development of the Gemma 3 family of models."

My question is: The paper’s key scalability claim relies entirely on the word "variant."

How different is this "variant" from the method proposed in the paper (Figure 3)?

Does it still involve "asking" with a complex prompt containing multiple criteria (informativeness, format, safety)? Or has it been reduced to a small component within a larger, more complex filtering pipeline?

If this "variant" is significantly different from the original method, the generalizability of the paper’s main conclusions (drawn from T5-800M) to the scale of Gemma 3 becomes far less certain.


3. Prompt Sensitivity
The entire magic of ASK-LLM is encapsulated in the prompt shown in Figure 3. This prompt itself is a complex, multi-dimensional instruction that requires the model to simultaneously judge "informational signals," "format," "knowledge," and "harmful content."

My question is: How sensitive are the results to the wording of this prompt?

What would happen if I removed the criterion "contain usable knowledge" and only kept "well-formatted"?

If I only asked "whether it contains harmful content," would it function as a safety filter?

The paper provides no ablation study on this newly introduced, complex "prompt hyperparameter."


In summary, these questions do not negate the merits of the paper, but they are the answers I would most want to know when considering deploying this method in practical production or conducting follow-up research.

---

> ### Author Response · Authors · 2025-12-02
>
> Thank you for your thoughtful and in-depth review. We address your questions below:
>
> 1. **Cost:** To answer your question, we mined our experiment logs for the accelerator-hour costs of scoring and training on C4. Note that our jobs ran at a low priority on a shared compute system and were preempted frequently. We use the accelerator duty cycle to estimate the time that it was in use, but this is an approximation because training and inference have different bottlenecks and can even occur on different hardware (for this data, we restrict our comparison to jobs on the same hardware type). We report the average cost over 30 training runs.
>
> | Scoring Model | Accelerator-Hr |
> | - | - |
> | T5-XXL | 49 |
> | T5-XL | 10 |
> | T5-Large | 1.7 |
> | T5-Base | 0.76 |
> | T5-Small | 0.24 |
>
> | Training Model | Accelerator-Hr |
> | - | - |
> | T5-Large | 24 |
> | T5-Small | 9.3 |
>
> **Cost-benefit analysis:** Figures 1 and 4 suggest that, conservatively, a 44% reduction is possible when training T5-L on data selected by T5-XL. This corresponds to 56\% $\times 24 + 10 \approx 23.44$ accelerator-hours with AskLLM compared to 24 accelerator-hours without AskLLM, with the benefit that the scores can be reused. Of course, these numbers are approximate, but the takeaway is that the costs are reasonable - especially as cheaper scoring models still have some benefit. We also note that AskLLM is cheaper than some techniques that are already used at scale (e.g., the logit-level distillation used in Gemma 3).
>
> 2. **Gemma 3 variant:** Unfortunately, we cannot disclose many details. We do still “ask” an LLM to judge quality based on a complex prompt with multiple criteria and broad discretion to decide what “high quality” means. Your pipeline questions are partially answered by the Gemma 3 paper, which states that AskLLM is used in conjunction with “filtering techniques that reduce the risk of unwanted or unsafe utterances and remove certain personal information and other sensitive data” as well as eval decontamination. AskLLM is also used for data re-weighting rather than for filtering.
>
> 3. **Prompt sensitivity:** We did not tune the prompt much, partly because we did not observe large differences from prompt changes during our development process. We do not discuss this in the paper, but one can cheaply estimate the prompt sensitivity via score correlations from different prompts (similar to Figure 7). High correlation (~0.9) is a sufficient but not necessary condition for similar downstream performance. Two prompts can achieve similar results even if they disagree on many points, because the downstream performance is an aggregate property of the training dataset selected by the LLM-judge.
>
> Of course, dramatically changing the selection criteria can affect performance. If we only kept “well-formatted,” we would likely pass more nicely-formatted but meaningless data (tables of numbers, Lorem-Ipsum, XML tags, etc). But it is not clear whether this difference would affect enough of the training set to harm downstream evals. If we only asked about harmful content, we would indeed get a safety filter - though the judge’s notion of safety may not be particularly well-aligned.
>
> **Weaknesses:** It is reasonable to expect quality, noise, and diversity to play different roles as the scale of the model changes. However, the literature contains some evidence that we can accurately estimate performance gains on large models using ablations on small models. For example, the authors of *“Scaling Laws for Optimal Data Mixtures”* use experiments on models of size (106M, 275M, 616M) to accurately predict performance at the (2.3B, 8B) scale. It is not reported how far this extrapolation can be extended.
>
> Regarding data cleanliness, it is true that AskLLM would be unable to filter much from a dataset that had already been processed by a strongly-correlated / aligned filter. However, there are many ways to use a quality score that go beyond the curation we perform in this paper. In general, the research community has moved from filtering by a single score in 2023/2024 to strategies that aggregate multiple scores and involve more intricate data mixing techniques in 2025. For example, the Gemma 3 whitepaper and *“Scaling Laws for Optimal Data Mixtures”* re-weight the data according to quality signals. We are happy to add a discussion of recent works to the paper (e.g., *QuRating*, *Datacomp-LM*, *QuaDMix* from Bytedance, *CLIMB* from Nvidia, etc). While there has been considerable recent work in this direction, we still feel that our large-scale comparisons, AskLLM methodology, and problems / comparisons between scoring strategies remain relevant (as many recent algorithms are score-consumers).
>
> Once again, thank you for taking the time to read our paper and for the insightful questions. We understand that further discussion is not possible due to the recent OpenReview bug / anonymization leak, but we do hope that you feel we have addressed your concerns.

---

### Official Review · Reviewer_rgfd · 2025-10-31

**Soundness:** 3
**Presentation:** 4
**Contribution:** 3
**Rating:** 8
**Confidence:** 3

**Summary:**

This paper studies data-efficient pretraining of large language models (LLMs) by systematically evaluating two data selection strategies integrated into the pretraining pipeline.The proposed ASK-LLM method uses an instruction-tuned model to score each pretraining sample via prompting for quality judgment, selecting data with high predicted informativeness and linguistic coherence. In contrast, DENSITY estimates sample representativeness using embedding-space density and selects a diverse subset that maximizes coverage of latent topics. Both methods are applied before model training as a data filtering stage, followed by standard T5 pretraining on the selected subsets under a fixed compute budget. The study compares 22 sampling schemes on subsets of the C4 corpus using T5-small and T5-large, and evaluates downstream on 111 tasks. Results show that ASK-LLM substantially improves data efficiency—matching or surpassing full-data training with 10–60% of data and converging up to 70% faster—while DENSITY performs best among coverage-based baselines. Random sampling remains competitive but inferior to the LLM-guided quality filtering approach.

**Strengths:**

- The paper studies how different data sampling strategies influence the efficiency of LLM pretraining under a clearly defined and reproducible setup.
- It conducts extensive and controlled experiments, covering 22 sampling strategies, over 200 pretraining runs, and 1,100 downstream fine-tunings under a fixed-compute regime.
- The methodology is clearly framed through a contrast between quality-driven (ASK-LLM) and coverage-driven (DENSITY) sampling, offering a structured perspective on data selection principles.
- Empirical results are consistent across model scales and sampling ratios: ASK-LLM yields larger gains in low-data regimes, while DENSITY maintains stable coverage performance, supported by convergence and ablation analyses.
- The experiments include explicit reporting of compute budgets, dataset proportions, and evaluation metrics, allowing clear interpretation and comparison of the data-efficiency results.

**Weaknesses:**

- The ASK-LLM sampler requires one inference per training example, which introduces notable computational cost. While the authors mention this can be amortized across multiple runs, the initial scoring overhead may affect scalability to larger datasets.
- The baselines do not include comparisons with more advanced recent methods such as Harnessing Diversity for Important Data Selection in Pretraining Large Language Models (ICLR 2025).
- All experiments are conducted on encoder–decoder (T5) models using text-only corpora, and the generalization of the findings to decoder-only or multimodal architectures has not been evaluated.

**Questions:**

- Did the authors experiment with alternative prompt formulations or response schemes for ASK-LLM—for example, using scalar or graded scoring instead of binary yes/no judgments? If not, how confident are they that the binary formulation captures sufficient quality signal?
- How sensitive are the ASK-LLM filtering results to the choice of the LLM judge? The paper suggests that different Flan-T5 judge sizes yield similar filtering behavior—could the authors clarify where this consistency arises from?

---

> ### Author Response · Authors · 2025-12-02
>
> Thank you for your positive review - we address weaknesses and questions below.
>
> 1. **Inference costs:** We mined our experiment logs to estimate the accelerator-hour costs for scoring the C4 dataset and training on C4. Note that our jobs ran at low priority on a shared cluster and were frequently pre-empted. We use the average accelerator duty cycle to estimate the fraction of time that the accelerator was in use by our job, but want to note that training and inference have different bottlenecks and can even occur on different hardware (for the data presented here, we restrict our comparison to jobs on the same hardware). For training cost numbers, we report the average over 30 runs.
>
> | Scoring Model | Accelerator-Hr |
> | - | - |
> | T5-XXL | 49 |
> | T5-XL | 10 |
> | T5-Large | 1.7 |
> | T5-Base | 0.76 |
> | T5-Small | 0.24 |
>
> | Training Model | Accelerator-Hr |
> | - | - |
> | T5-Large | 24 |
> | T5-Small | 9.3 |
>
> **Cost-benefit analysis:** Here is an example cost-benefit analysis using AskLLM-T5-XL to train T5-L. Figures 1 and 4 suggest that, conservatively, a 44% reduction in training horizon is possible with no performance loss. This corresponds to a cost of  56\% $\times 24 + 10 \approx 23.44$ accelerator-hours for AskLLM-sampled training compared to a cost of 24 accelerator-hours without AskLLM, with the benefit that the AskLLM scores can be reused by future runs. Of course, these values are approximate but the takeaway is that the scoring costs are reasonable (especially for cheaper scoring models).
>
> 2. **Baselines, multimodality:** We are unable to add further experiments with recent techniques, as each baseline requires ~10 pre-training runs and >50 fine-tuning runs. However, we are happy to discuss this paper as well as other recent work (e.g., *QuRating*, *Datacomp-LM*, *QuaDMix* from Bytedance, *CLIMB* from Nvidia, and *“Scaling Laws for Optimal Data Mixtures”* which are all very recent).
>
> Our LLM scoring framework can accommodate multimodal inputs, but we expect this direction to bring many new research questions (e.g., how would one treat an example with excellent text but poor images? How about data with poor alignment between images / audio / video / text?). Therefore, we leave this exciting direction for future work.
>
> 3. **Decoder-only architectures.** We can report that a variation of AskLLM was used successfully in the training of the Gemma 3 series of models (briefly mentioned in the paper). The main differences lie in the prompt and in how the score is used (data re-weighting rather than filtering). Unfortunately, it is not possible for us to provide ablations with decoder-only architectures that compare different methods of quality ranking in a controlled setting.
>
> 4. **Prompting schemes.** We have explored different ways to structure the prompt, though unfortunately not in a setting that permits direct comparisons with the numbers in the paper. Qualitatively, we observe that LLMs exhibit poor calibration when asked to output a numeric score directly, resulting in noisy labels for borderline data and degenerate distributions. Likert rubrics seem to require require highly-detailed, case-by-case grading criteria to avoid instruction-following failures. We do not observe this issue with token-probability P(yes) scores, possibly for the same reason that one can improve the calibration of LLM raters by transforming scalar-output tasks into token-probability tasks (e.g., see *“Self-Evaluation Improves Selective Generation in Large Language Models”*).
>
> 5. **LLM judge sensitivity:** One interesting trend is that despite similar downstream performance, the different AskLLM judges are not strongly correlated on individual score pairs (see Figure 7). We believe this occurs because the downstream performance of a model trained on sampled data is an aggregate property of the training dataset. Two judges can achieve similar performance even if they disagree on many points, as we observe in our experiments.
>
> Appendix H contains a detailed analysis of different LLM judges. H.3 and H.4 contain qualitative examples of agreement and disagreement between the T5 judges, and the general rule is that larger model sizes can distinguish quality in more nuanced ways (e.g., advertisement copy is selected by small judges but not by large ones). We examine the topic affinity for various LLM judges in Appendix G (Figure 11). While we observe near-identical affinity for perplexity-based filters, AskLLM affinities differ significantly among the different LLM judges (even within the Flan-T5 family).
>
>
> Once again, thank you for your positive review and for taking the time to read our paper. We will add a summarized form of this discussion to the main text. We understand that further discussion is not possible due to the recent OpenReview bug / anonymization leak. Even still, we hope that you feel we have addressed your concerns.

---

### Official Review · Reviewer_WkEw · 2025-10-31

**Soundness:** 3
**Presentation:** 3
**Contribution:** 3
**Rating:** 6
**Confidence:** 2

**Summary:**

This paper investigates data-efficient pre-training methods for LLMs to reduce high training costs. It contrasts two primary data selection strategies: data quality and data coverage. The authors introduce two novel methods to represent these strategies:
1. ASK-LLM: A quality-based sampler that uses an instruction-tuned LLM to perform a zero-shot, prompt-based assessment of a data point's quality.
2. DENSITY: A coverage-based sampler that selects a diverse set of examples by modeling the data distribution in the feature space.

In a large-scale experiment testing 22 different curation techniques on T5-style models, ASK-LLM was the top performer. The key finding is that while coverage-based samplers typically match the performance of training on the full dataset, the quality-based ASK-LLM consistently outperforms full-data training.

**Strengths:**

1. The paper's conclusions are supported by a massive-scale experiment. The authors tested 22 different data curation techniques , involving 220 pre-training runs and 1,100 distinct fine-tuning runs , all evaluated on 111 downstream tasks. This comprehensive and costly analysis provides high-confidence evidence for their claims.
2. The primary strength is the clear finding that quality-based filtering (ASK-LLM) can train models that outperform models trained on the entire dataset, even when using as little as 10% of the data. This is a significant result because it shows data curation is not just about matching full-data performance more efficiently, but about achieving a new, higher performance ceiling.
3. The proposed ASK-LLM sampler is a novel and intuitive method that leverages an LLM's reasoning to score data quality. This approach is shown to be superior to traditional perplexity filtering because it captures a different, more valuable signal. The paper demonstrates that ASK-LLM scores have "almost no ranking correlation" with perplexity scores and that it correctly identifies high-quality, niche-topic examples that perplexity filtering wrongly penalizes.

**Weaknesses:**

1. The ASK-LLM method requires one full LLM inference pass for every single example in the training corpus. For a dataset like C4 (364M examples), this is a massive computational cost that could rival the cost of training itself. While the authors argue this cost is "amortized" over many training runs, this presents a very high, practical barrier to entry.

**Questions:**

1. The paper's evaluation relies entirely on post-finetuning performance across 111 tasks . How do you expect the ASK-LLM curation to affect the zero-shot and few-shot reasoning capabilities of the pre-trained models? Is there a risk that filtering for "informative" data (as defined by the prompt) optimizes for finetuning performance at the expense of the broad, general-purpose in-context learning abilities that are a key benefit of pre-training?
2. The ASK-LLM sampler requires a full inference pass over the entire dataset, which is a significant, up-front computational cost. While you argue this cost is amortized, could you provide a more concrete cost-benefit analysis? For example, using the T5-Large model, how many compute-hours did the ASK-LLM (XL) scoring pass on the full C4 dataset take?

---

> ### Author Response · Authors · 2025-12-02
>
> Thank you for taking the time to review our paper. Below, we address the weaknesses and questions.
>
> 1. **Regarding pretraining and fine-tuning:** Not all of our evals are post-finetuning; we do look at perplexity evals of the pretrained model. In particular, we measure perplexity on a high-quality held-out dataset that contains a broad set of knowledge-heavy passages. This is fairly standard practice for evaluating data curation methods, as it is well-documented that training dataset perplexity is a flawed way to measure model quality when ablating the data (see Appendix F.2 for details).
>
> In general, you are correct that data curation carries a risk of capability / knowledge loss (e.g., a model that has never seen grammatically-incorrect English may have difficulty if asked to role play as someone with limited English proficiency). This problem is unavoidable, as demonstrated by the “No free lunch” theorems in *“Data pruning and neural scaling laws: fundamental limitations of score-based algorithms”* (TMLR 2023). The intuition is that one can always construct an adversarial eval out of exactly the points that were discarded by the sampling algorithm.
>
> However, we do not see much evidence that AskLLM is removing useful knowledge or capabilities. For example, AskLLM improves perplexity on our HQ eval dataset, suggesting good preservation of useful knowledge. We would also like to note that T5 finetuning does not really add new capabilities but rather just makes it possible to elicit abilities that were instilled by pretraining (e.g., by training to respond in the proper format to multiple-choice questions or instructions). If AskLLM does well in this kind of post-finetuning eval, it implies that we did not remove the capabilities needed by that evals from the pretraining checkpoint. The fact that we see good performance across a diverse set of post-finetuning evals strongly suggests that AskLLM preserves many abilities.
>
> 2. **On inference vs. training cost:** We agree that this is an important point in practice. We mined our experiment logs for accelerator-hour estimates for some of our training and inference runs, obtaining the following numbers. Note that our jobs were run at a low (batch) priority on a shared compute system where pre-emption frequently occurs. To get a reasonable estimate, we use the total average accelerator duty cycle to estimate the fraction of time that the accelerator was in use by our job, but it is important to remember that training and inference have different bottlenecks (i.e., the accelerator duty cycle would not be 100% even for high-priority jobs that run without preemption).
>
> With these caveats in mind, here are cost estimates for scoring the C4 dataset and for training on C4.
>
> | Scoring Model | Accelerator-Hr |
> | - | - |
> | T5-XXL | 49 |
> | T5-XL | 10 |
> | T5-Large | 1.7 |
> | T5-Base | 0.76 |
> | T5-Small | 0.24 |
>
> | Training Model | Accelerator-Hr |
> | - | - |
> | T5-Large | 24 |
> | T5-Small | 9.3 |
>
> For the training cost numbers, we report the average cost over 30 training runs. Note that our experiments were conducted in the iso-compute setting, as described in Section 4, where the goal is to maximize performance at a fixed training cost. To do a cost-benefit analysis, we need to examine the amount by which we can shorten the training process without performance loss.
>
> **Example cost-benefit analysis:** Figures 1 and 4 suggest that, conservatively, a 44% reduction in training horizon is possible when training T5-L on data selected by T5-XL. This corresponds to a cost of 56\% $\times 24 + 10 \approx 23.44$ accelerator-hours for AskLLM-sampled training compared to a cost of 24 accelerator-hours without AskLLM, with the benefit that the AskLLM scores can be reused by future runs.
>
>
> Of course, the numbers used in this calculation are approximate, but the takeaway is that the costs are reasonable. This is particularly true when we consider that there are benefits even when using T5-Base as the scoring model (which is much cheaper).
>
> We are happy to add a summarized form of this discussion to the main text. We also noticed some formatting problems in the paper while responding to this review (mostly in the Appendix), which we will fix in the revision.
>
> We understand that further discussion is not possible due to the recent OpenReview bug / anonymization leak. Even still, we hope that you feel this response addresses your concerns and we thank you for your reviewing efforts.

---

### Official Review · Reviewer_1AMo · 2025-11-02

**Soundness:** 4
**Presentation:** 4
**Contribution:** 3
**Rating:** 6
**Confidence:** 4

**Summary:**

- This paper addresses the critical challenge of efficient data selection in LLM pre-training. The core research question investigates the trade-offs between data selection strategies based on data quality versus those based on coverage.
- To explore this, the authors propose two novel data curation techniques: (1) ASK-LLM, a quality-based sampler that leverages the zero-shot reasoning of an instruction-tuned LLM to directly score the informativeness of each training example, and (2) DENSITY, a coverage-based sampler that uses kernel density estimation on data embeddings to select a diverse subset of the data.
- The authors conduct an extensive empirical study by pre-training T5-style models (60M and 800M parameters) on various subsets of the C4 dataset curated by 22 different methods.

**Strengths:**

- The motivation of this paper is clear: selecting high-quality and high-coverage data for efficient pretraining.
- While model-based data selection has been explored, the use of an instruction-tuned LLM's explicit, prompt-based reasoning to assess the quality of individual pre-training samples is a novel and powerful concept.
- The experimental results are highly insightful, demonstrating that (1) compared to other model-based methods (e.g., PPL), the proposed ASK-LLM leverages the reasoning and context ability from the instruct-tuned LLM for long-tail high-quality data selection. (2) With the improvements of the scoring model, there are increasingly beneficial effects as the scaling of the to-be-trained model. (Perplexity filters do not seem to exhibit such trends.)
- The paper is well-written, well-organized, and easy to follow.

**Weaknesses:**

- This work conducts all experiments on the T5 series encoder-decoder LLMs. As of late 2025, the vast majority of state-of-the-art open-source research and applications are centered on decoder-only LLMs, like Qwen/Llama/Gemma/Mistral. While the authors acknowledge this limitation, the absence of any experiments on a decoder-only architecture makes it difficult to assess the direct generalizability of the findings.
- The paper argues that the high inference cost of ASK-LLM is amortized over multiple training runs. However, scoring a trillion-token dataset is much expensive. The paper would benefit from a more concrete cost analysis. For example, a simple table estimating the TPU/GPU-hours required to score the full C4 dataset with Flan-T5-XL versus the compute saved from the accelerated T5-Large training run would provide a much clearer picture of the return on investment.

**Questions:**

Same as the weaknesses.

---

> ### Author Response · Authors · 2025-12-02
>
> Thank you for your review of our paper. Below, we address the weaknesses and questions.
>
> **Re: decoder-only architectures.** We can report that a variation of AskLLM was used successfully in the training of the Gemma 3 series of models (this is mentioned in the paper, albeit briefly). Regrettably, we are not able to release detailed ablations with decoder-only architectures that compare different methods of quality ranking in a controlled setting.
>
> **Re: scoring cost.** This is a valid point and is certainly a concern in practice. To answer your question, we mined our experiment logs to get accelerator-hour estimates of some of our training and inference runs. We are able to report the following numbers, with the caveat that our jobs were run at a low (batch) priority on a shared compute system and were preempted frequently. We use the total average accelerator duty cycle to estimate the fraction of time that the accelerator was in use by our job, but we want to note that this is an approximation because training and inference have different bottlenecks (i.e., the accelerator duty cycle will not be 100% even for high-priority jobs that run without preemption).
>
>
> With those disclaimers in mind, here are cost estimates for scoring the C4 dataset and for training on C4.
>
>
> | Scoring Model | Accelerator-Hr |
> | - | - |
> | T5-XXL | 49 |
> | T5-XL | 10 |
> | T5-Large | 1.7 |
> | T5-Base | 0.76 |
> | T5-Small | 0.24 |
>
>
>
>
> | Training Model | Accelerator-Hr |
> | - | - |
> | T5-Large | 24 |
> | T5-Small | 9.3 |
>
> For the training cost numbers, we report the average cost over 30 training runs. Note that our experiments were conducted in the iso-compute setting, as described in Section 4, where the goal is to maximize performance at a fixed number of training tokens. To estimate the cost savings, we need to instead look at the amount by which we can reduce the training cost while maintaining neutral performance.
>
> **Cost-benefit analysis:** Here is an example cost-benefit analysis using AskLLM-T5-XL to train T5-L. Figures 1 and 4 suggest that, conservatively, a 44% reduction in training horizon is possible with no performance loss. This corresponds to a cost of 56\% $\times 24 + 10 \approx 23.44$ accelerator-hours for AskLLM-sampled training compared to a cost of 24 accelerator-hours without AskLLM, with the benefit that the AskLLM scores can be reused by future runs.
>
> Of course, the numbers used in this calculation are approximate, but the takeaway is that the costs are reasonable. This is particularly true when we consider that there are benefits even when using T5-Base as the scoring model (which is considerably cheaper). We would also like to note that AskLLM is already cheaper than several techniques that are already used at scale (such as the logit-level teacher-student distillation that was used in Gemma 3, which we discuss in Appendix C1).
>
> We are happy to add the tables and cost analysis to the main text. We also noticed some formatting problems in the paper while responding to this review (mostly in the Appendix), which we will fix in the revision.
>
> We understand that further discussion is not possible due to the recent OpenReview bug / anonymization leak. Even still, we hope that you feel we have addressed your concerns and we sincerely thank you for your time in reviewing our paper.

---

### Meta-Review · Area_Chair_jA42 · 2025-12-16

**Summary:**

This paper presents work on data-efficient pre-training for large language models (LLMs). It analyzed the trade-offs between data quality estimation and coverage/diversity-based selection. The work introduces AskLLM, which uses instruction-tuned LLMs to assess data quality in a zero-shot manner, and density sampling to improve coverage in feature space. Extensive pre-training and downstream evaluations on T5-style models show that density-based methods recover full-data performance, while AskLLM consistently outperforms full-data training using only a fraction of the data and achieves faster convergence. The study provides clear empirical insights into practical data curation strategies for efficient LLM pre-training.

This paper received reviewing comments from five domain experts. They acknowledged that the research problem of this paper is meaningful. The proposed method is insightful. Although there are some questions or concerns about the current form, its contribution makes it meet the acceptance line. All reviewers are positive. The area chair checked their comments and this paper, and agreed with the reviewers. Therefore, this work is recommended for acceptance.

**Reviewer Concerns:**

Reviewers' concerns include the computational cost of the proposed method, the language model used in the experiment, the sensitivity of the prompt, and other questions about the method's mechanism. By checking the rebuttal, except for the fact that no other larger or latest models were introduced in the experiment (the authors claimed that the variant of the proposed method has been applied), all other concerns were resolved.

**Reviewer Scores:**

**Reviewer 1AMo**
The concerns of this reviewer mainly include the computational cost of the proposed method and a more comprehensive evaluation with other LLMs. If the reviewer had been able to participate fully in the discussion, the score would remain the same or increase, since the question of the computational cost was answered. The question of the evaluation with other LLMs was partially answered, although there is no detailed supplemented result.

 **Reviewer WkEw**
The concerns mainly include the lack of ability brought about by the proposed method and the computational cost. The concerns would be addressed if the reviewer participated fully in the discussion, with a high probability. The score would be increased.

 **Reviewer rgfd**
The concerns are reflected in the computational cost, the used LLMs, prompts, and the used judgment model. Except for the used LLMs, the other questions were solved by checking the rebuttal. Therefore, the reviewer would still be positive about the work.

 **Reviewer 72fC**
The concerns are reflected in the computational cost, the used LLMs, prompts, and more details of the data filter mechanism of the proposed method under some cases. The first three main questions have been answered. For the last question, it is also important. However, the qualitative explanation may not be very convincing. Therefore, this reviewer may not change the score if he/she can participate fully in the discussion.

 **Reviewer THC1**
The concerns are reflected in computational cost, the used LLMs, and method defects (self-reinforcing bubble). The first and second questions were addressed (i.e., a variation of AskLLM was used). For the problem of the self-reinforcing bubble, the rebuttal gave some explanations and suggestions, but without solid results. If the reviewer can participate fully in the discussion, he/she would keep support but not increase the score further.

---

### Decision · Program_Chairs · 2026-01-26

Accept (Poster)